# A novel triptolide analog downregulates NF-κB and induces mitochondrial apoptosis pathways in human pancreatic cancer

Qiaomu Tian[1†], Peng Zhang[2†], Yihan Wang[1], Youhui Si[1], Dengping Yin[1], Christopher R Weber[3], Melissa L Fishel[4], Karen E Pollok[4], Bo Qiu[2], Fei Xiao[2], Anita S Chong[1]*

[1]Department of Surgery, The University of Chicago, Chicago, United States; [2]Cinkate Pharmaceutical Corp, ZhangJiang District, Shanghai, China; [3]Department of Pathology, The University of Chicago, Chicago, United States; [4]Department of Pediatrics, Indiana University, Indianapolis, United States

*For correspondence: achong@bsd.uchicago.edu

†These authors contributed equally to this work

**Abstract** Pancreatic cancer is the seventh leading cause of cancer-related death worldwide, and despite advancements in disease management, the 5-year survival rate stands at only 12%. Triptolides have potent anti-tumor activity against different types of cancers, including pancreatic cancer, however poor solubility and toxicity limit their translation into clinical use. We synthesized a novel pro-drug of triptolide, (*E*)–19-[(1'-benzoyloxy-1'-phenyl)-methylidene]-Triptolide (CK21), which was formulated into an emulsion for in vitro and in vivo testing in rats and mice, and used human pancreatic cancer cell lines and patient-derived pancreatic tumor organoids. A time-course transcriptomic profiling of tumor organoids treated with CK21 in vitro was conducted to define its mechanism of action, as well as transcriptomic profiling at a single time point post-CK21 administration in vivo. Intravenous administration of emulsified CK21 resulted in the stable release of triptolide, and potent anti-proliferative effects on human pancreatic cancer cell lines and patient-derived pancreatic tumor organoids in vitro, and with minimal toxicity in vivo. Time course transcriptomic profiling of tumor organoids treated with CK21 in vitro revealed <10 differentially expressed genes (DEGs) at 3 hr and ~8,000 DEGs at 12 hr. Overall inhibition of general RNA transcription was observed, and Ingenuity pathway analysis together with functional cellular assays confirmed inhibition of the NF-κB pathway, increased oxidative phosphorylation and mitochondrial dysfunction, leading ultimately to increased reactive oxygen species (ROS) production, reduced B-cell-lymphoma protein 2 (BCL2) expression, and mitochondrial-mediated tumor cell apoptosis. Thus, CK21 is a novel pro-drug of triptolide that exerts potent anti-proliferative effects on human pancreatic tumors by inhibiting the NF-κB pathway, leading ultimately to mitochondrial-mediated tumor cell apoptosis.

## Editor's evaluation

Tian et al. describe a novel modified version of the pro-drug triptolide, CK21, and provide evidence for its improved pharmacokinetics and its safety and efficacy in multiple xenograft models of pancreatic cancer. The authors performed transcriptomic analysis upon CK21 treatment which revealed that downregulation of NF-κB and mitochondrial dysfunction induce apoptosis and therefore lead to tumor regression. Downregulation of NF-κB and induction of apoptosis was then validated in vitro and in vivo. These findings have potential clinical significance as the efficacy of CK21 in preclinical PDAC models is compelling.

**eLife digest** Pancreatic cancer is a major cause of cancer-related deaths worldwide, with only 12% of patients surviving for five years after diagnosis. Individuals generally experience few symptoms of the disease in the early stages and are often diagnosed once the cancer has already spread to other parts of the body. By this point, options for treatment are limited.

A molecule known as triptolide has been shown to kill breast, lung, pancreatic and other types of cancer cells. However, triptolide is toxic to humans and other animals, making it unsuitable for use in patients. One way to make drugs safer without compromising their beneficial effects is to modify their molecular structure. By formulating triptolide into an emulsion – a mixture of liquids allowing it to dissolve – Tian, Zhang et al. synthesized a new analogue called CK21.

Experiments showed that CK21 inhibited the growth of human pancreatic cancer cells grown in a laboratory including cells grown in artificial organs similar to the pancreas, known as pancreatic tumor organoids. Furthermore, CK21 killed large tumors in mice pancreases with very few side effects, suggesting the structural modification of triptolide increased safety of the drug.

To better understand how CK21 works, Tian, Zhang et al. examined the genes that were induced in the pancreatic tumor organoids at various time points after treatment with the drug. This revealed that CK21 switched off genes involved in the NF-κB cell signaling pathway, which regulates how cells grow and respond to stress. In turn, it triggered programmed cell death, killing the tumor cells in a controlled manner.

The findings suggest that CK21 could be a promising candidate for treating pancreatic cancer. In the future, clinical trials will be required to establish whether CK21 is a safe and effective therapy for humans.

## Introduction

Pancreatic cancer is the seventh leading cause of cancer -related deaths globally and the third leading in the United States, and has the lowest 5 -year survival rate among all the cancers (*DeSantis et al., 2019*). Pancreatic ductal adenocarcinoma accounts for >90% of all pancreatic cancer cases, and poor outcomes have been attributed to late diagnoses when the cancer is at advance stages (*Kamisawa et al., 2016*), where the majority of cases are accompanied with distant metastasis (*Yachida et al., 2010*; *Sohn et al., 2000*) and when most patients are not eligible for resection (*Bilimoria et al., 2007*). Fluorouracil, and gemcitabine are FDA approved as adjuvant chemotherapy after pancreatic cancer resection (*Oettle et al., 2013*), FOLFIRINOX, Abraxane with gemcitabine represent first-line chemotherapy for patients with metastatic pancreatic cancer (*Burris et al., 1997*; *Von Hoff et al., 2013*; *Conroy et al., 2011*). For patients with resectable disease followed by adjuvant chemotherapy, anticipated median overall survival Is 54.4 months, however, for patients with advanced unresectable disease, the survival benefit with multiagent chemotherapy is only 2–6 months (*Kamisawa et al., 2016*).

The Chinese herb, *Tripterygium wilfordii* hook F (Thunder God vine), has anti-inflammatory, immunosuppressive, contraceptive, and anti-tumor activities, and has been used for centuries as traditional Chinese medicine for treating rheumatoid arthritis and lupus. In 1972, Morris et al. extracted triptolide from *T. wilfordii* and characterized it as a structurally unique diterpene triepoxide, with potential anti-leukemic properties (*Kupchan et al., 1972*). Subsequently, triptolide was shown to have anti-tumor effects in pre-clinical mouse models of breast cancer (*He et al., 2020*; *Li et al., 2014b*), cholangio-carcinoma (*Liu, 2011*), osteosarcoma (*Jiang et al., 2017*), lung cancer (*Reno et al., 2015*; *Song et al., 2017*), acute myeloid leukemia (*Carter et al., 2012*; *Carter et al., 2006*), ovarian cancer (*Hu et al., 2016*; *Zhao et al., 2012*), prostate cancer (*Huang et al., 2012*), gastric cancer (*Yang et al., 2003*), colon cancer (*Wang et al., 2009*), and pancreatic cancer (*Chugh et al., 2012*; *Wang et al., 2012*). Multiple mechanisms have been proposed for triptolide-induced antitumor activity, including inhibition of NF-κB (*Lee et al., 2002*), and HSP70 (*Phillips et al., 2007*). Notably, *Titov et al., 2011* reported that triptolide binds covalently to human XPB (ERCC3) and inhibits its DNA-dependent ATPase activity, leading to the inhibition of RNA polymerase II-mediated transcription and nucleotide excision repair. However, it is unclear how this non-specific inhibition of an essential transcription factor could exert selectivity against tumors.

While triptolide is a promising anti-cancer drug, poor solubility and toxicity have limited its clinical development, and a number of analogs of triptolide have been developed for improved clinical performance (*Noel et al., 2019*; *Tong et al., 2021*). In Phase I clinical studies, a soluble analog PG490-88/F60008 (*Kitzen et al., 2009*) resulted in significant toxicity and had high interindividual variability in pharmacokinetic studies, thus stopping further development. Minnelide (*Greeno et al., 2015*) is another analog with superior solubility and potent anti-tumor 1activity in multiple preclinical cancer models. Phase I clinical trial (ClinicalTrials.gov Identifier: NCT03129139) showed significant activity in highly refractory metastatic pancreatic cancer, and it is currently in a Phase II open label trial (ClinicalTrials.gov ID NCT03117920).

In this study, we synthesized a novel pro-drug of triptolide, CK21, by decorating the C-19 with a C-C double bond to generate (*E*)–19-[(1'-benzoyloxy-1'-phenyl)-methylidene]-Triptolide, formulated it into an emulsion, and investigated its efficacy and mode of action. We report that CK21 inhibited the in vitro proliferation of multiple pancreatic cancer cell lines, was effective at eliminating large pancreatic tumors in heterotopic and orthotopic xenograft animal models with minimal toxicity, and confirmed the efficacy of CK21 against multiple patient-derived pancreatic tumor organoids in vitro and in vivo. We performed transcriptome analysis on the pancreatic organoid response to CK21 in vitro, and on the in vivo response of pancreatic tumors to CK21. We identified that CK21 reducing overall transcription, inhibited the NF-κB pathway, induced mitochondria dysfunction, and ultimately, mitochondrial-mediated apoptosis was identified as the likely mechanism for the anti-tumor activity of CK21.

## Results

### Novel modified triptolide, CK21, show improved pharmacokinetics

We designed a new modification strategy to triptolide to generate CK21, by decorating the C-19 with a C-C double bond to generate (*E*)–19-[(1'-benzoyloxy-1'-phenyl)-methylidene]-Triptolide (*Figure 1a*). Briefly, a mixture of triptolide (1.8 g, 5 mmol) with anhydrous tetrahydrofuran (250 mL) was kept at –25 °C~–20 °C under nitrogen protection. Benzoyl chloride (1.05 mL, 7.5 mmol) and Lithium 2,2,6,6-tetramethylpiperidine in tetrahydrofuran/toluene (7.5 mL, 2.0 M, 15 mmol) were then added dropwise to produce an intermediate compound, IM464. After 1 hr, addition of benzoyl chloride and lithium 2,2,6,6-tetramethylpiperidine was repeated, and the reaction was quenched by adding aqueous sodium carbonate (6%). Following concentration under reduced pressure, the crude product was separated and purified by silica gel chromatography, and the target product collected and further recrystallized in methylene chloride/hexane to obtain CK21 that was used in the in vitro studies. Using $^1$H NMR, $^{13}$C NMR and mass spectrometry, we confirmed the structure of CK21, and the absolute configuration of CK21 was established by single crystal X-ray diffraction (*Figure 1b*). We then formulated CK21 with medium chain triglycerides, phospholipids, glycerol, and DSPE-MPEG2000 (*Figure 1c*) to produce a CK21 emulsion (*Figure 1d*) that was used in the in vivo studies.

To examine the conversion of CK21 into triptolide in vivo, and to establish pharmacokinetics and to avoid toxicity, we intravenously administrated 3 mg/kg or 1.5 mg/kg CK21 into Sprague Dawley male or female rats, and the concentration of CK21 and triptolide in the plasma quantified. CK21 had a $T_{1/2}$ of 1.3 hr and 0.225 hr for male and female rats respectively. Released triptolide reached $T_{max}$ at 0.25 and 0.75 hr with a $C_{max}$ of 78.3 and 81.9 nM respectively for male and female rats. A stable release of triptolide 30 nM to 80 nM was observed for up to 2 hours and was undetectable after 4 hours (*Figure 1e*), which we hypothesize may mitigate the toxicity observed with other triptolide derivatives, which exhibit a spike release (*Kitzen et al., 2009*). The maximum tolerated dose (MTD) of CK21 was 3 mg/kg/dose for female rats and 6 mg/kg/dose for male rats (*Figure 1—source data 1*). Finally, we observed that in vitro incubation of the human pancreatic cancer cell lines, AsPC-1 and Panc-1, with CK21 at 5–100 nM for 24, 48, and 72 hr resulted in a dose-and time-dependent inhibition of cell proliferation (*Figure 1f*). When co-cultured with primary human fibroblast for 72 hr, CK21 exhibited significant toxicity only at 500 nM or higher (*Figure 1—figure supplement 1*).

A comparison of CK21 and triptolide (TP) revealed that they had similar IC50 (nM) when tested in vitro using a cell viability assay with different cancer cell lines (*Figure 1—source data 2*). However, the in vivo toxicity of TP in mice was significantly higher than CK21 in vivo (*Figure 2—figure supplement 1*).

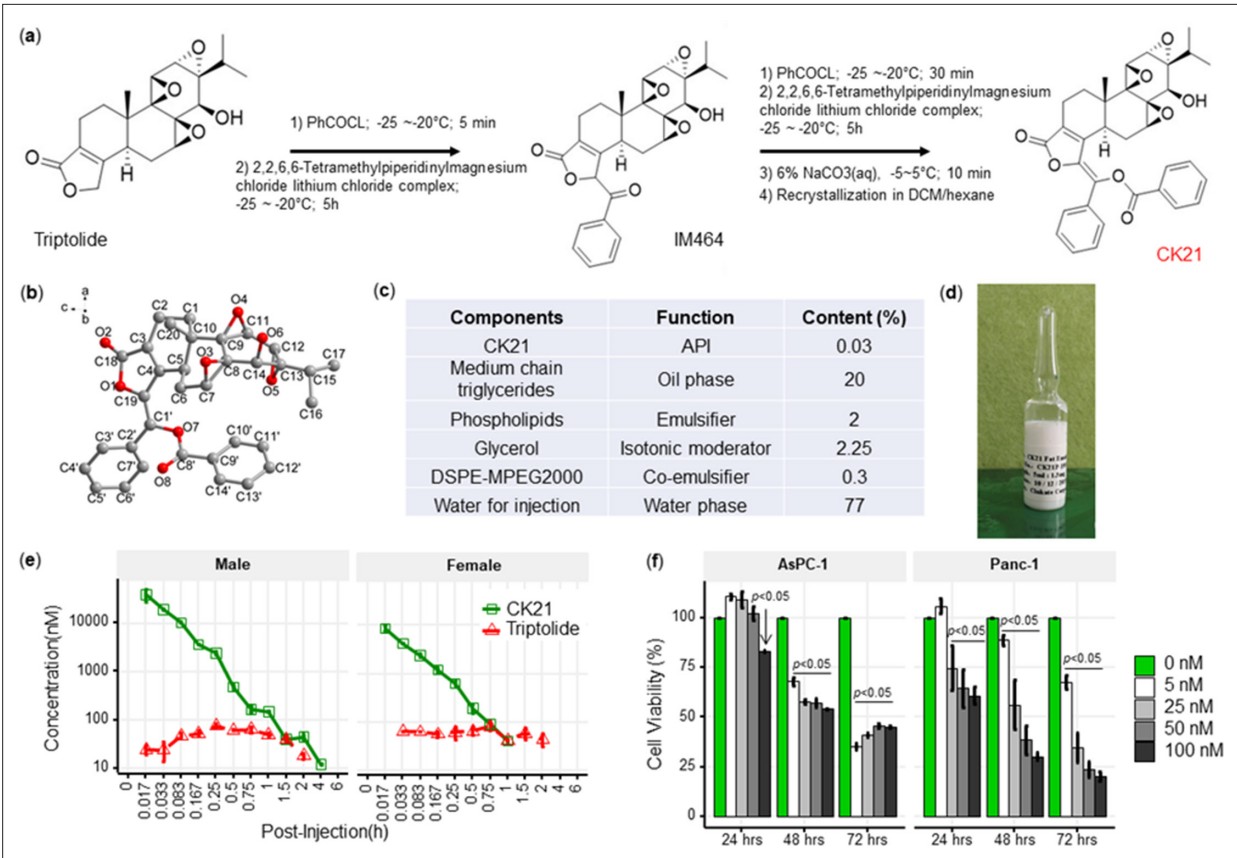

**Figure 1.** CK21 exhibits a stable release of triptolide in vivo. (**a**) Synthesis of compound CK21 as white solid after recrystallization in a mixed organic solvent. Compound structure was characterized by H-NMR, C-NMR, and HR-MS. (**b**) Thermal ellipsoid model illustrating the crystal structure of CK21; carbon atoms were shown in gray, and oxygen atoms in red. Hydrogen atoms were omitted for clarity (**c**) Composition and putative function in the CK21 fat emulsion. (**d**) Macroscopic image of the final emulsion product of CK21. (**e**) In vivo administration of CK21 into SD rats (3 rats per group) converted into triptolide. CK21 was injected intravenously into female (1.5 mg/kg) and male (3 mg/kg) rats, and the concentration of CK21 and triptolide in the plasma was quantified. For samples ≥4 hours, no CK21 or triptolide was detected. (**f**) CK21 inhibited the proliferation of human pancreatic cancer cell lines. Data presented in all the graphs are mean ± standard error. Statistical analysis: Two-way ANOVA (repeated measures) with post-hoc comparison of the means was conducted for (**f**).

The online version of this article includes the following source data and figure supplement(s) for figure 1:

**Source data 1.** Safety profile of CK21.

**Source data 2.** IC50 (µM) of triptolide (TP) or CK21 for different cancer cell lines in an in vitro cell viability assay.

**Figure supplement 1.** In vitro viability assay of primary human fibroblasts cocultured with CK21 at the indicated concentrations for 72 hours.

## CK21 inhibits AsPC-1 and Panc-1 proliferation in vitro and tumor growth in vivo

To evaluate the efficacy of CK21 pro-drug in vivo, we developed a xenograft model where AsPC-1 tumors were subcutaneously implanted into female nude mice (*Figure 2a*). Daily treatment with CK21 at all doses tested (1.25, 2.5, 3 and 5 mg/kg) significantly inhibited AsPC-1 tumor growth (*Figure 2c*). Higher dosages of CK21 at 3 mg/kg or 5 mg/kg daily eliminated the tumor after 28 days of treatment (*Figure 2b*). After 28 days of CK21 treatment, no mice from 3 mg/kg or 5 mg/kg groups demonstrated tumor relapse during the subsequent 6 -month follow-up observation (*Figure 2—figure supplement 2*).

No significant weight loss was detected when female mice were treated with ≤3 mg/kg CK21, compared to the control (no treatment) group (*Figure 2d*). In contrast, mice exhibited severe weight loss with 5 mg/kg CK21. To further confirm the lack of toxicity of CK21 (3 mg/kg), we performed H&E staining on the kidney, liver, and pancreas of mice after 28 days treatment. We did not observe any evidence of toxicity, as the kidney, liver, and pancreas tissues appeared normal after 28 days of

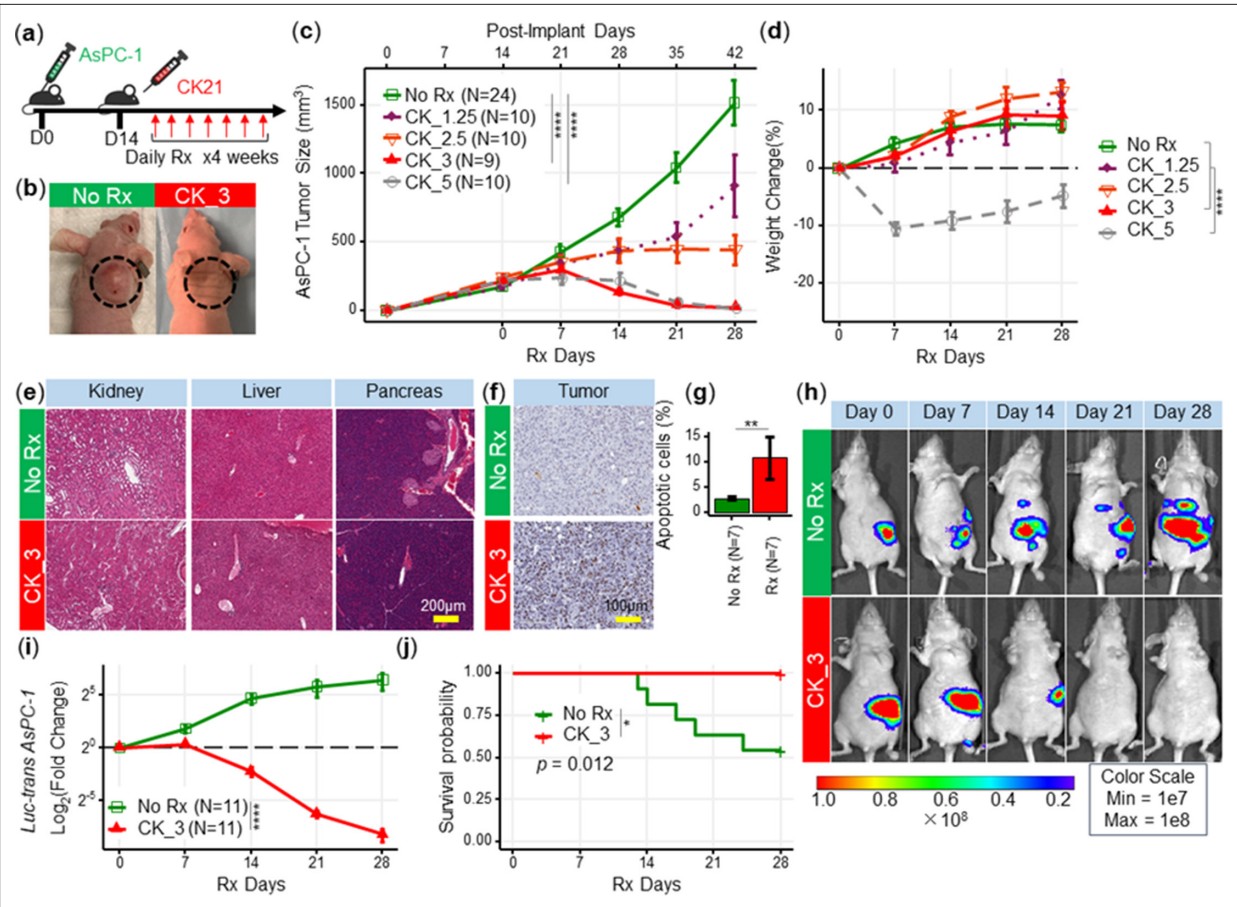

**Figure 2.** CK21 shows efficacy and minimal toxicity at 3 mg/kg in different in vivo animal models. (**a**) Scheme of in vivo efficacy studies. Human pancreatic cancer cell line, AsPc-1, was implanted into nude mice and CK21 treatment was initiated ~14 days later and administered daily for 4 weeks. (**b**) Macroscopic images of tumor-bearing nude mice after receiving CK21 or blank emulsion after 4 weeks treatment. (**c**) AsPC-1 tumor volume after subcutaneous implantation and CK21 or gemcitabine treatment. (**d**) Weight change of nude mice bearing AsPC-1 and receiving CK21. (**e**) H&E staining of mice organ tissues after CK21 treatment. (**f**) TUNEL staining of tumor tissue and (**g**) percentage of apoptotic cells in AsPC-1 tumor after 2 weeks CK21. (**h**) Bioluminescence images of nude mice bearing intra-pancreatic AsPC-1 and receiving CK21. Color scheme represents the intensity of luminescence reflecting tumor size in each mouse. Mice with higher initial tumor burden was placed into CK21 group, and those with lower initial tumor burden into control group. (**i**) Fold change of the luminescence intensity of the nude mice bearing intra-pancreatic AsPC-1. (**j**) Survival curve of mice with orthotopic AsPC-1 tumors receiving CK21 treatment. In all the figures, post-implant days are days after tumor implantation and post-Rx days are days after receiving CK21 treatment (doses indicated as mg/kg). Data presented in all the graphs are mean ± standard error (some error bars are too small to be visible). Statistical analysis: Two-way ANOVA (not repeated measures) with post-hoc comparison of the means of each data set was conducted for all the line graphs except (**i**); For survival curve, Log-rank (Mantel-Cox) test was applied. (* $P<0.05$, ** $P<0.01$, *** $P<0.001$, **** $P<0.0001$).

The online version of this article includes the following figure supplement(s) for figure 2:

**Figure supplement 1.** Survival curve of mice receiving CK21 at 5 mg/kg or triptolide (TP) at 0.25 mg/kg.

**Figure supplement 2.** AsPC-1 subcutaneous tumors showed no tumor relapse after treated with CK21 at 5 or 3 mg/kg.

**Figure supplement 3.** CK21 inhibited growth of Panc-1 tumors in a subcutaneous xenograft model.

**Figure supplement 4.** Male mice with AsPC-1 tumors respond to CK21.

CK21 treatment (*Figure 2e*); in contrast, after 14 days of CK21 treatment, AsPC-1 tumors showed a 5-fivefold increase of TUNEL-positive staining compared to the no Rx group (*Figure 2f, g*). Thus, we concluded that CK21 given at 3 mg/kg daily exhibited high efficacy and minimal toxicity, and this dose was employed for the remaining of study. In a second subcutaneous xenograft model with the Panc-1 tumor cell line, 3 mg/kg daily of CK21 also resulted in significant inhibition of tumor growth (*Figure 2—figure supplement 3*).

Orthotopic tumor mouse models are generally preferred over heterotopic subcutaneously located pancreatic tumors because they offer tissue site-specific pathology, allow studies of metastasis, and

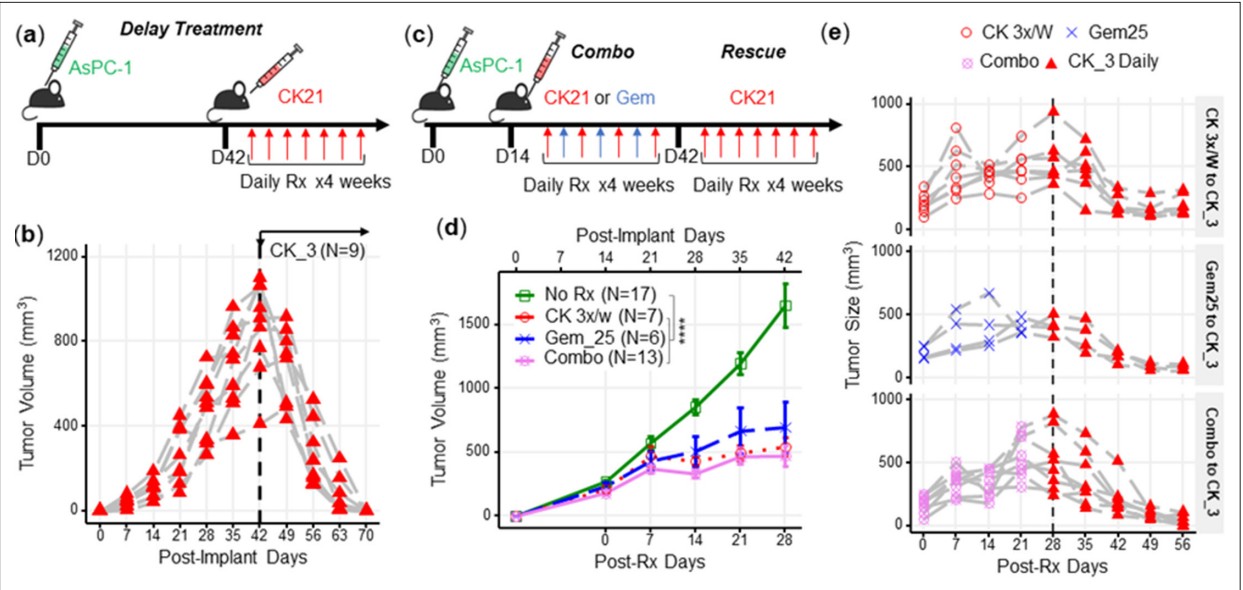

**Figure 3.** CK21 of 3 mg/kg daily shows efficacy in delay therapy and rescues mice that failed in synergistic therapy. (**a**) Scheme of delayed therapy. Mice received CK21 at 3 mg/kg daily for 4 weeks, starting on day 42 post-tumor innoculation. (**b**) Tumor volume during delayed CK21 therapy. (**c**) Scheme of combination (Combo) and rescue therapy. Mice receive CK21 3 mg/kg (3 X/week; Mo, We, Fr), gemcitabine at 25 mg/kg (3 X/week; Tu, Th, Sa), or both. (**d**) Tumor size during the Combo therapy of CK21. (**e**) Mice which failed at CK21 or gemcitabine or Combo therapy from (**c–d**) were then rescued by switching to CK21 at 3 mg/kg daily, and tumor size monitored. Post-implant days are days after tumor implantation. Post-Rx days are days after receiving drug treatment. Data presented in (**d**) are mean ± standard error. Statistical analysis: Two-way ANOVA (not repeated measures) with post-hoc comparison of the means of each data set was conducted for (**d**), (**\*\*\*\*** *P*<0.0001). Each line in (**b**) and (**e**) represents a single mouse.

are deemed more clinically relevant (**Qiu and Su, 2013**). while the development of pancreatic tumors expressing luciferase/fluorescent proteins has facilitated the longitudinal monitoring of orthotopically located pancreatic tumors (**Shannon et al., 2015**). We next evaluated the efficacy of CK21 in an orthotropic xenograft model, using luciferase-transfected AsPC-1 implanted into the pancreas of nude mice and allowing the tumor to develop for 1–2 weeks before initiating CK21 treatment. The presence and size of the tumor were monitored weekly by quantifying the bioluminescence intensity (**Figure 2h**), and overall, a 10–15-fold reduction in bioluminescence intensity was observed in mice that received CK21 compared to untreated controls (**Figure 2i**). In addition, no mice died in the CK21 treatment group, whereas 5 out of 11 animals were sacrificed in the no Rx group due to the large tumor size (**Figure 2j**). Finally, we noted that while most of the untreated mice develop metastatic disease by the end of the experiment (**Figure 2h**), the CK21 treated mice did not. After 4 weeks of treatment, mice were monitored up to 3 months. All mice relapsed eventually in contrast to subcutaneous AsPC-1 tumors.

## Delayed CK21 therapy inhibits growth of tumors that escaped earlier therapies

The mortality of pancreatic tumors is often due to late detection when the tumor is at an advanced stage. To evaluate the efficacy of CK21 against late-stage tumors, CK21 treatment was initiated only after subcutaneous AsPC-1 tumors reached a large size of ~900 mm² (**Figure 3a**). Despite this delay in the initiation of treatment, CK21 was able to completely reduce the size of AsPC-1 tumors after 28 days of treatment, with all mice showing a significant response (**Figure 3b**).

Gemcitabine is a standard of care medication for pancreatic cancer in the clinic (**Kamisawa et al., 2016**), therefore we next tested whether gemcitabine in combination with CK21 might offer improved efficacy. We treated mice for 4 weeks with suboptimal doses of CK21 (3 mg/kg, 3 days/wk) and gemcitabine (25 mg/kg, 3 days/wk), with each drug given on alternate days to avoid toxicity (**Figure 3c**). The combination therapy did not show improved inhibition of AsPC-1 growth compared to CK21 monotherapy (**Figure 3d**) and failed to induce complete regression of AsPC-1 tumors. In mice where tumors were detectable after 28 days treatment with CK21 or gemcitabine monotherapy,

or combination therapy, we tested whether switching to CK21 (3 mg/kg) daily treatment (*Figure 3e*) was able to induce tumor regression. We observed that irrespective of whether mice failed CK21 (3 x/wk) or gemcitabine monotherapy, or combination therapy, switching to daily CK21 monotherapy for 28 days induced significant tumor regression (*Figure 3e*).

## Transcriptome analysis of patient-derived organoids revealed early down-regulation of DDIT4 and XBP1 by CK21

It is now recognized that 3-D patient-derived organoids offer a better recapitulation of the heterogeneous, architectural, morphologic and genetic features of patient pancreatic tumor, compared to long-term established 2-D monolayer cell lines (*Weeber et al., 2017*; *Huang et al., 2015*; *Seino et al., 2018*; *Boj et al., 2015*). We therefore investigated four organoids derived from different pancreatic cancer patients (*Romero-Calvo et al., 2019*), UC12-0118-8, U049MAI, U123SOK, and U123M15-T, and tested the susceptibility to CK21 in vitro and in vivo. Details of the origin, mutations of these organoids were described in *Figure 4—source data 1*. We observed that 72 hours of in vitro incubation with CK21 (25 nM) significantly inhibited UC12-0118-8, U049MAI, and U123SOK growth, and CK21 (50 nM) significantly inhibited proliferation of all four organoids (*Figure 4a*). In addition, we were able to propagate U049MAI as a slow-growing subcutaneous tumor in nude mice. Treatment with CK21 (3 mg/kg, daily) for 28 days, also significantly reduced U049MAI tumor growth compared to the untreated control group (*Figure 4b*).

Because pancreatic tumor organoids better preserve the genetic signatures than pancreatic tumor cell lines, we performed a time-course RNA-seq of U049MAI and U123M15-T treated with CK21 for 3, 6, 9, and 12 hoursr. We hypothesized that these early time points might reveal the initiating mechanism of action that result ultimately in the control of tumor growth; indeed, the number of differentially expressed genes (DEGs) significantly increased with prolonged CK21 treatment, from less than 10 DEGs at 3 hr up to 8,000 DEGs at 12 hr (*Figure 4c* and *Figure 4—figure supplement 1*). We identified the genes that were differentially expressed at early time points and continuously upregulated or downregulated at later time points (*Figure 4d*). We confirmed with qPCR, of a significant downregulation of DDIT4, MYC, XBP1 and XIAP, as well as a significant upregulation of POLR2A, GADD45 and VAMP1 (*Figure 4e*). We also performed transcriptome analysis on the AsPC-1 tumor, orthotopically implanted in the pancreas for 7 days and then treated by CK21 for three days. Notably, CK21 induced similar DEG expression profiles as in vitro treated organoids, with downregulated DDIT4 and XBP1, as well as upregulated POLR2A (*Figure 4g*).

DDIT4 was one of the genes consistently and strongly downregulated by CK21 in both organoids and AsPC-1, with significant effects observed as early as 3 hoursr of CK21 treatment in vitro and at day 3 in vivo. At the protein level, we also observed a significant decrease of DDIT4 expression after CK21 treatment of 24 hoursr (*Figure 4—figure supplement 2*). Interestingly, DDIT4 has been identified as a prognosis marker and highly expressed in pancreatic tumors (*Pinto et al., 2017*), thus prompting the investigation into whether DDIT4 inhibition might be the triggering mechanism of action and thus serve as a predictive biomarker for CK21 sensitivity. However, knock-down of DDIT4 in Panc-1 only induced very modest in vitro susceptibility to CK21, and the overexpression of DDIT4 in AsPC-1 didn't result a difference to CK21 response (*Figure 4—figure supplement 3*). Furthermore, in two mouse pancreatic tumor cell lines derived from genetically modified KC or KPC mice that were only modestly sensitive to CK21 treatment (*Figure 4—figure supplement 4*), DDIT4 as well as other early responder genes showed strong alterations in expression profiles comparable to tumors that were more sensitive to CK21 (*Figure 4—figure supplement 5*). Therefore, these early responder genes are not likely to be essential mediators leading to tumor susceptibility to CK21.

## Ingenuity pathway analysis of patient-derived organoids reveal down-regulation of the NF-κB signaling pathway by CK21

At the later timepoint of 12 hr after CK21 treatment, both U049MAI and U123M15-T had over 8,000 DEGs compared to the no Rx group (*Figure 5a, b*). We then used Ingenuity pathway analysis (IPA, Qiagen) on the DEGs to identify the major molecular and cellular functions that were significantly affected by CK21 treatment (*Figure 5c*). First, CK21 treatment was predicted to inhibit RNA and DNA transcription, expression of RNA, and transactivation of RNA transcription in both organoids; this observation corroborates a previous report on the ability of triptolide to inhibit RNA transcription

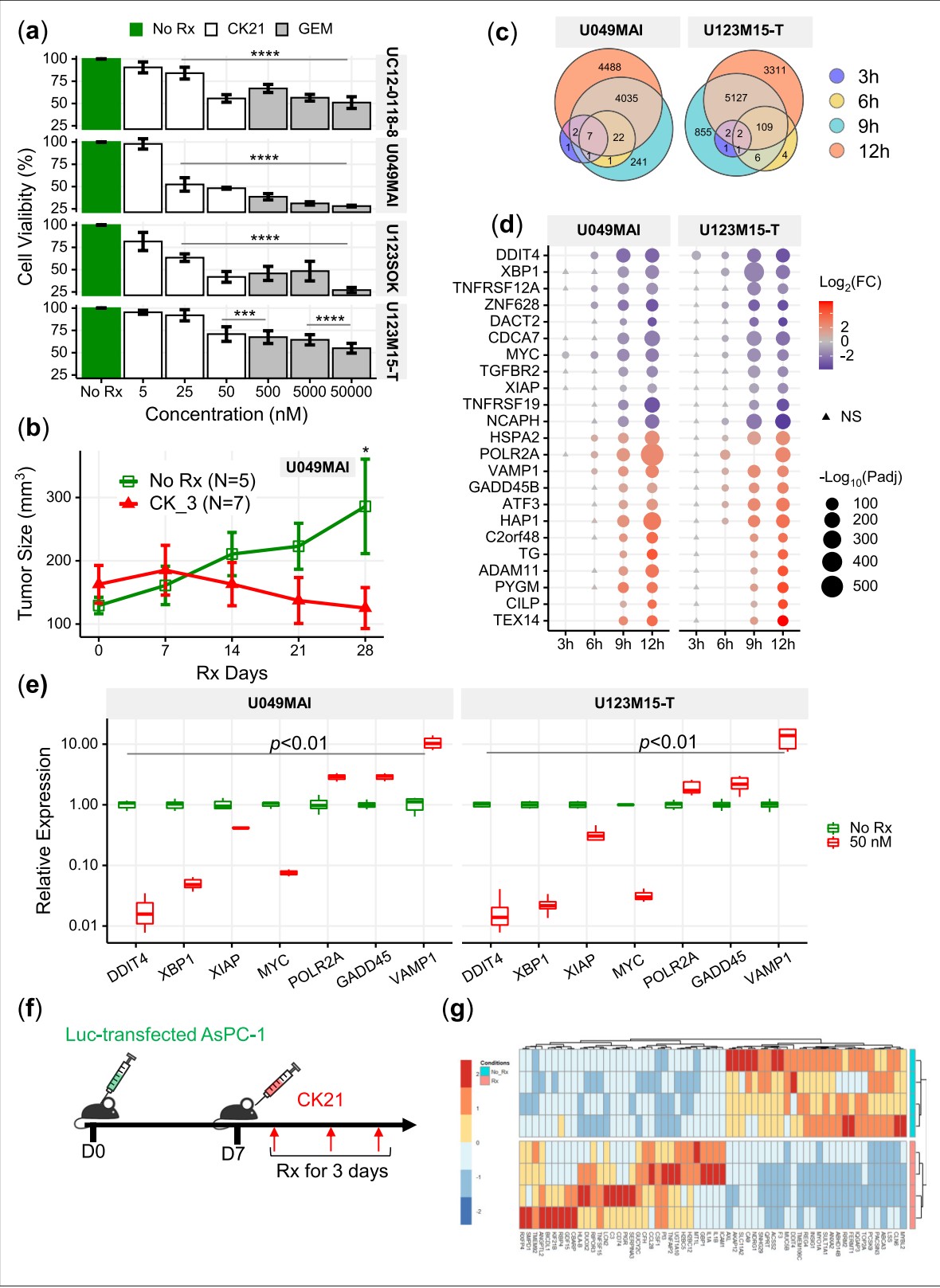

**Figure 4.** Transcriptome analysis of patient-derived pancreatic tumor organoids and AsPC-1 after CK21 treatment. (**a**) In vitro culture of different organoids with escalating concentrations of CK21 for 72 hours. Gemcitabine was included as a positive control. (**b**) U049MAI tumor size in nude mice during CK21 treatment. (**c**) Co-expression Venn diagram of differentially expressed genes that were significantly different with CK21 treatment. Size of the circles reflect the total number of differentiate expressed genes (transformed using log2(n+1)). (**d**) Genes of interest showing consistent up or down

*Figure 4 continued on next page*

*Figure 4 continued*

regulation as treatment time increased. Fold change is color coded where red is upregulation, blue is down regulation. Circle presents the genes had an adjusted *P*-value <0.05, and triangle presents the genes had an adjusted p*P*-value >0.05. Size of the circle represents the adjusted p values. (**e**) RT-qPCR analysis of gene expression in tumor organoids after CK21 treatment for 24 hours. (**f**) Scheme of RNA seq using in vivo orthotropic AsPC-1 model. (**g**) Heatmaps of top statistically significant differentially expressed genes in AsPC-1 tumors after treatment with CK21 for three days. Statistical analysis: Two-way ANOVA (not repeated measures) with post-hoc comparison of the means of each drug dose was compared to No Rx controls for (**a**). Line indicates the doses that resulted in significant reduction in viability by CK21 or gemcitabine. Two-way ANOVA with post-hoc comparison of the means of each time point was conducted for (**b**), Multiple t tests were conducted for (**e**) (** *P*<0.01, *** *P*<0.001, **** *P*<0.0001).

The online version of this article includes the following source data and figure supplement(s) for figure 4:

**Source data 1.** Essential information on the pancreatic tumor organoids used in this study.

**Figure supplement 1.** Volcano plots highlighting differentially expressed genes by U049MAI and U123m15-T respectively after 3 hours, 6 hours, 9 hours, and 12 hours of CK21 (50 nM) treatment.

**Figure supplement 2.** DDIT4 expression was significantly reduced after CK21 treatment.

**Figure supplement 3.** DDIT4 knockdown or overexpression didn't impact the sensitivty of tumor cell lines to CK21 treatment.

**Figure supplement 4.** Tumor size of KC-6141 and KPC-961 after subcutaneous implantation in B6 or B6 X129.

**Figure supplement 5.** RT-qPCR analysis of differentially expressed genes by two mice pancreatic tumor cell lines after CK21 treatment at 50 nM for 24 hours.

(*Titov et al., 2011*). In addition, DEGs induced by CK21 were enriched for inhibition of cell proliferation and cell survival, and for inducing apoptosis and tumor cell necrosis. These observations collectively are consistent with TUNEL-positive staining of ASPC-1 with CK21 treatment in vivo, and support the conclusion that induction of cell apoptosis is the likely mechanism for the anti-tumor activity of CK21.

We used IPA pathway enrichment analysis to further identify the canonical signaling/metabolic pathways regulated by CK21 that might lead to tumor cell apoptosis (*Figure 5d, e*). Interestingly, in both organoids, EIF2 signaling, oxidative phosphorylation and mitochondrial dysfunction were the major pathways highly upregulated by CK21, whereas the NF-κB, TGF-ß and telomerase signaling pathways were significantly downregulated at the 12 hr treatment timepoint. In addition, at 9 hour timepoint, NF-κB was already significantly downregulated and oxidative phosphorylation as well as EIF2 signaling pathway were significantly upregulated (*Figure 5—figure supplement 1*). In vivo, Aspc-1 orthotopic tumors showed upregulation of DNA damage checkpoint regulation (*Figure 5— figure supplement 2*), which also is an indicator of tumor apoptosis. Collectively, these observations suggest that CK21 may be inhibiting NF-κB activity and inducing mitochondrial-mediated tumor cell apoptosis.

## CK21 inhibits expression of NF-κB p65 and translocation to nuclei

NF-κB plays a major role in the regulation of immune, inflammatory response and cell proliferation (*Park and Hong, 2016*). In normal cells, NF-κB is activated by appropriate stimuli and then returns to its inactive state. In tumor cells, particularly in pancreatic cancer cells, NF-κB becomes constitutively activated and has an anti-apoptotic function (*Liptay et al., 2003*; *Dolcet et al., 2005*). After 12 hr treatment with CK21, the genes (CHUK, IKBKB and RELA) encoding the key regulators of the NF-κB pathway, IKKα, IKKβ and p65, were significantly downregulated in both organoids (*Figure 6a*).

To confirm the transcriptional findings that CK21 downregulates the NF-κB pathway, we stained the nuclei and p65 of AsPC-1 and Panc-1 with different fluorophores to visually determine their cellular location; similarity in the spatial localization between p65 and nuclei represents the translocation of NF-κB to nuclei (*Figure 6b*). In the no Rx group, p65 staining had a high similarity with nuclei staining, corresponding with constitutive nuclear localization of NF-κB in pancreatic cancer cells. After treatment with CK21 for 24 or 48 hours, both cell lines exhibited significantly lower expression of p65, consistent with RNA-seq analysis (*Figure 6c*). In addition, we observed reduced similarity of p65 and nuclei, indicating significantly reduced translocation of NF-κB to the nuclei in the presence of CK21 (*Figure 6d, e*). Taken together, the data demonstrate that CK21 inhibits NF-κB expression and translocation, which we hypothesize results in increased susceptibility tumor cell apoptosis.

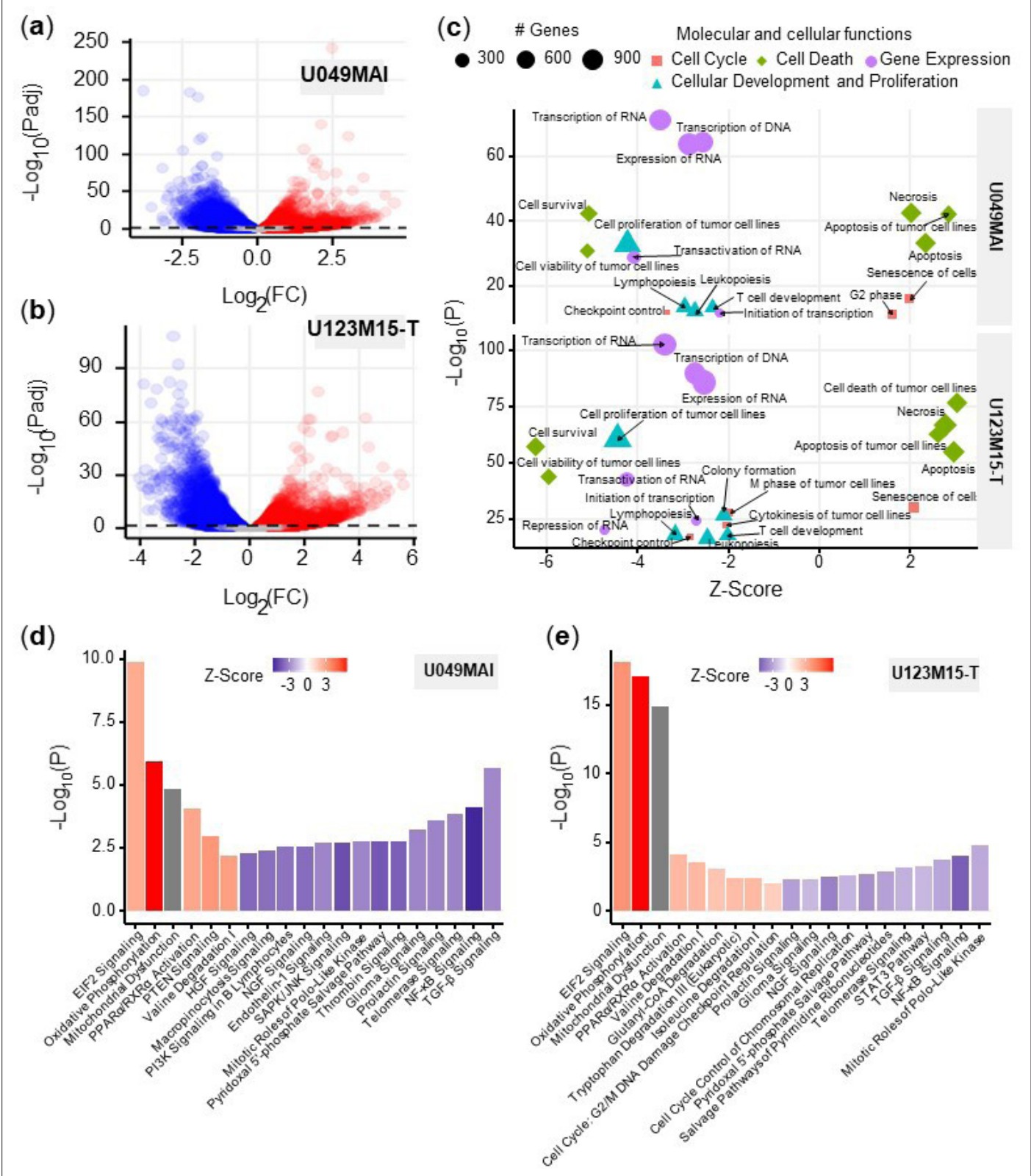

**Figure 5.** Bioinformatic analysis of the effect of CK21 on patient-derived pancreatic tumor organoids. (**a**) Volcano plots of differentially expressed genes in (**a**) U049MAI and (**b**) U123M15-T after 12 hours CK21 treatment (50 nM). Significance cutoff was s $P<0.05$. Upregulation was colored as red, and downregulation was colored as blue. (**c**) Enrichment of molecular and cellular functions in U049MAI and U123M15-T after CK21 treatment. Size represents gene numbers. Color and shape represent functional groups. Z-score represents the confidence of the prediction, where positive value

*Figure 5 continued on next page*

*Figure 5 continued*

means upregulation and negative value means downregulation. Canonical pathway enrichment in (**d**) U049MAI and (**e**) U123M15-T after treatment with CK21 at 50 nM. Color represent Z-score where red means upregulation and blue means downregulation. Statistical analysis: Unpaired t-test was conducted for (**c**); Data presented in all the bar graphs are mean ± standard error.

The online version of this article includes the following figure supplement(s) for figure 5:

**Figure supplement 1.** Pathway enrichment of U049MAI and U123M15-T after treatment with CK21 (50 nM) for 9 hours.

**Figure supplement 2.** Pathway enrichment of orthotopic AsPC-1 tumors after treatment with CK21 (3 mg/kg) for 3 days.

## CK21 induces reactive oxidative species and mitochondrial mediated apoptosis

The expression of genes encoding five mitochondrial respiratory chain complexes were significantly increased in pancreatic tumor organoids treated with CK21 (*Figure 6f*), consistent with dysregulated mitochondrial function and increased susceptibility to mitochondrial-mediated apoptosis (*Márquez-Jurado et al., 2018*). Because mitochondrial -mediated apoptosis is often stimulated by oxidative stress, we first tested whether CK21 induced reactive oxidative species (ROS) in AsPC-1 and Panc-1 pancreatic tumor cell lines. In both cell lines, a trend towards an increase in ROS was observed as early as 8 hours after CK21 treatment, and a significant increase in ROS generation after 24 hours of culture with CK21 (*Figure 6g*). These observations raise the possibility that increased ROS production may trigger mitochondrial outer membrane permeabilization and release of pro-apoptotic mitochondrial proteins into the cytoplasm (*Márquez-Jurado et al., 2018*).

The B-cell-lymphoma protein 2 (BCL2) family of proteins also play critical roles in regulating the mitochondrial pathway of apoptosis, and BCL2 functions as a critical anti-apoptotic survival protein (*Redza-Dutordoir and Averill-Bates, 2016*). To test whether BCL2 protein is reduced in CK21-treated cells, we quantified BCL2 protein expression by Western blotting. We observed that BCL2 was significant decreased in both AsPC-1 and Panc-1 cell lines, and in U049MAI, after 24 hours of CK21 culture (*Figure 6h*).

Because most apoptotic pathways lead to the activation of cysteine-dependent aspartate-specific proteases, and ultimately to cleaved effector caspases such as caspases-3,–6 and –7 *Redza-Dutordoir and Averill-Bates, 2016*, we probed for cleaved caspase-3 in pancreatic tumors incubated with CK21. For Panc-1 and both pancreatic tumor organoids, cleaved caspase-3 was detected after 24 hours of culture with CK21 (*Figure 6i*) by Western blotting. We also confirmed increased caspase-3/7 in Panc-1 by flow cytometry (*Figure 6—figure supplement 1*). Interesting, cleaved caspase-3/7 was not detected in AsPC-1 after CK21 treatment, suggesting that apoptosis of these tumor cells may be explained by the involvement of other effector caspases or proteases. Collectively, these data point to CK21 downregulating the NF-kB pathway, promoting ROS production and mitochondrial-mediated tumor cell apoptosis.

## CK21 showed minimal immunosuppression in a spontaneous tumor rejection model

A number of studies have reported on the immunosuppressive activity of triptolide (*Chen, 2001*), thus raising the potential concern that CK21 may also inhibit the development of anti-tumor immune responses and prevent long-term tumor control. Indeed, although the analyses were conducted on CK21 treated tumor cells, IPA analysis indicated that CK21 inhibited lymphopoiesis, leukopoiesis and T cell development, consistent with potential immunosuppressive activity. To address this concern, we utilized a mouse KPC-960 pancreatic ductal-like tumor model derived from pancreatic tumors that spontaneously arose in KPC (*Kras*$^{G12D/+}$*Trp53*$^{R172H/+}$*Pdx1-Cre*) B6.129 mice (*Torres et al., 2013*; *Figure 7a*). Upon subcutaneous implantation into B6.129 immunocompetent hosts, KPC-960 grew to a maximum tumor size by day 7 and then approximately 70% KPC-960 tumors were spontaneously rejected by day 14–17 post-implantation (*Figure 7b*). This contrasted with tumor formation in similar B6.129 host in *Torres et al., 2013*; we speculate that rejection of the KPC-960 tumor may be driven increased number of passages that resulted in the accumulation of mutations resulting in antigenic drift. To test whether CK21 could prevent the spontaneous regression of KPC-960, CK21 (3 mg/kg daily) therapy was initiated on day 5 or 7 post-implantation. We observed no statistically significant inhibition of tumor regression when CK21 treatment was started on day 5 or 7 post-implantation

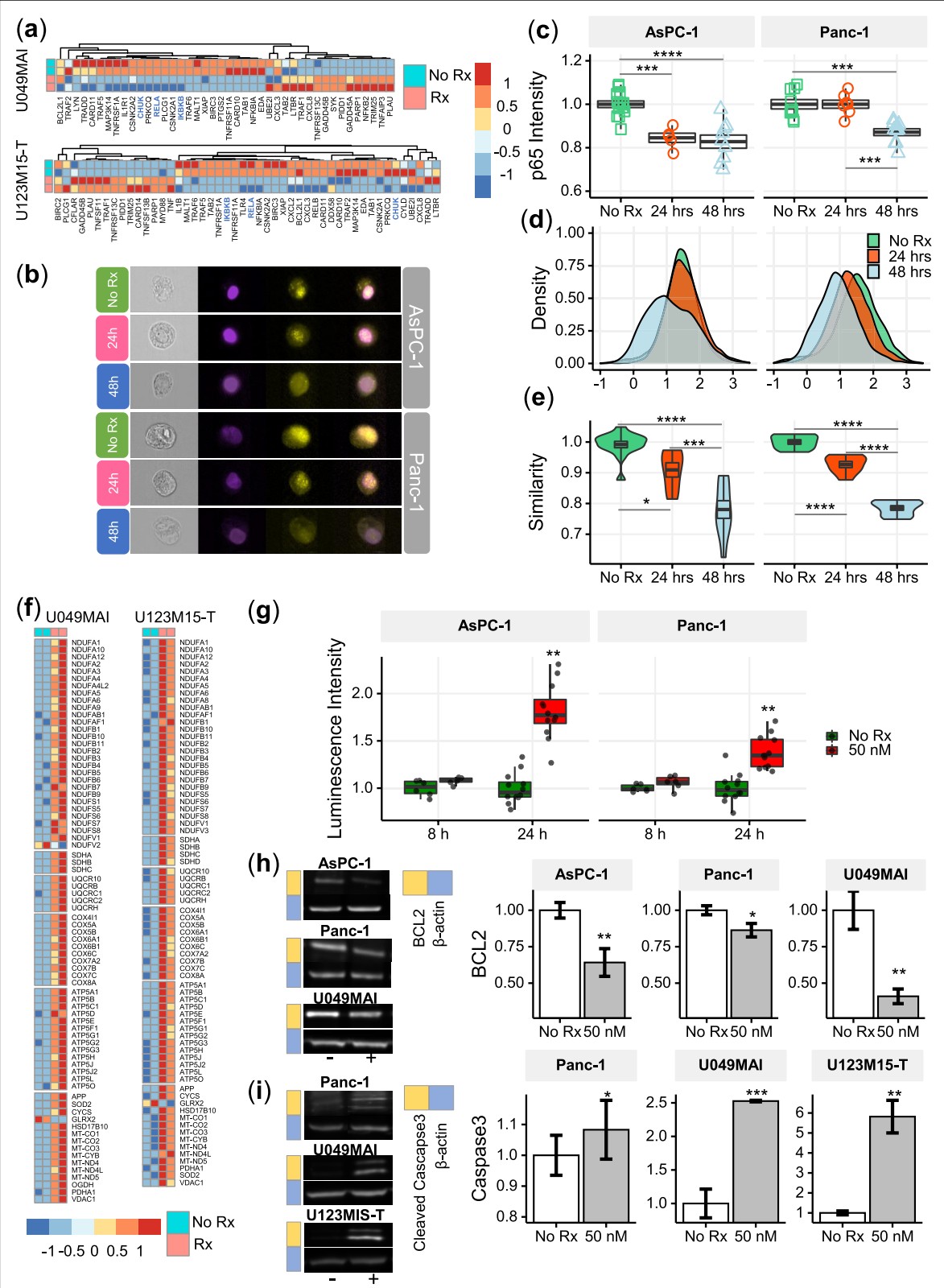

**Figure 6.** CK21 inhibits NF-κB activation and induces mitochondrial mediated apoptosis. (**a**) Heatmap of the relative expression of genes in the NF-κB pathway in U049MAI and U123M15-T after CK21 treatment. Genes are color coded where red means upregulated, and blue means downregulated. Only statistically significant genes are listed. (**b**) Representative p65 translocation images of AsPC-1 and Panc-1 after treated with CK21 at 50 nM. Nuclei stained as purple, p65 stained as yellow. (**c**) Relative p65 MFI of AsPC-1 and Panc-1 after CK21 (50 nM) treatment. (**d**) Density plots and (**e**) similarity

*Figure 6 continued on next page*

*Figure 6 continued*

scores of p65 for AsPC-1 and Panc-1. (**f**) Heatmaps of genes involved in oxidative phosphorylation of U049MAI and U123M15-T after CK21 treatment. (**g**) Reactive oxygen species generated after CK21 treatment (8 and 24 hoursr). Representative blotting images and quantification of (**h**) BCL2 expression and (**i**) cleaved caspase-3 at 24 hours after CK21 treatment. Statistical analysis: One-way ANOVA with post-hoc Tukey comparison of the means of each data set was conducted for (**c**), (**e**); Unpaired T test was conducted at different time points for (**g**), (**h**), (**i**). (* $P<0.05$, ** $P<0.01$, *** $P<0.001$, **** $P<0.0001$).

The online version of this article includes the following source data and figure supplement(s) for figure 6:

**Source data 1.** Full unedited gels.

**Figure supplement 1.** Flow plots illustrating active Caspase 3/7 expression in AsPC-1 and Panc-1 treated with CK21 (50 and 400 mM) for 24 hoursr.

**Figure supplement 2.** Key regulators in NF-kB canonical signaling pathway are significantly downregulated in (**a**) U049MAI and (**b**) U123M15-T after treatment with CK21 (50 nM) for 12 hoursr.

(*Figure 7c, d*) suggesting that the immunosuppressive activity of CK21 on established primary immune responses is minimal. We also implanted KPC-960 subcutaneously into nude mice but observed limited efficacy by CK21 when provided at 3 mg/kg/day (*Figure 7—figure supplement 1*). These observations suggest that host immunity is primary responsible for the rejection of KPC-960 tumors. The reason for KPC-960 resistance to CK21 is not known and is the subject of future investigations.

We next tested the possibility that CK21 may have inhibited the development of memory and recall anti-tumor responses that mediate the spontaneous rejection of secondary KPC-960 tumors. Mice that cleared these tumors were rested for 2 weeks without treatment and then challenged with a second KPC-960 tumor (*Figure 7a*); a more rapid tumor clearance was observed (*Figure 7e*). When CK21 treatment was initiated on day 3 of second tumor implantation, no significant change in the kinetics of tumor regression was observed compared to untreated controls (*Figure 7f*). In addition, mice that rejected the first KPC-960 tumors while receiving CK21 were rested and re-challenged with a second KPC-960 tumor. All the mice were able to reject the tumor comparably to those that did not receive CK21, (*Figure 7g*). These observations further demonstrate CK21 did not inhibit the development of memory or recall anti-tumor responses.

Finally, to evaluate the quality of tumor-specific T cells after CK21 treatment, we performed an ex vivo tumor killing assay. Splenocytes were harvested from untreated mice that had rejected tumors, or mice that had received CK21-treatment after 1° or 2° tumor implantation and cultured with KPC-960 or a control KPC-6141 tumor ex vivo (*Figure 7h*). Splenocytes from mice treated with CK21 exhibited comparable killing of KPC-960 as splenocytes from untreated mice (*Figure 7i*). Collectively, these data suggest that despite potent anti-tumor activity, CK21 was minimally immunosuppressive.

## Discussion

Toxicity is the key challenge for using triptolide and its derivatives as anti-tumor agents in the clinic. Hepatotoxicity, reproductive toxicity, and nephrotoxicity have been identified as the major side effects for triptolide (*Li et al., 2014a*). In addition, sex differences have been observed, where the female rats showed more toxicity under the same dosage of triptolide (*Liu et al., 2010*). Cytochrome P450s (CYP) is essential for the metabolism of triptolide and CYP3A2, a male-predominant form in rats, may contribute to the sex-related differences (*Xue et al., 2011*). Similar sex differences were also observed for CK21, where half the dose of CK21 in female rats had a similar triptolide exposure in plasma as male rats (*Figure 1e*), and the maximum tolerated dose of CK21 was 3 mg/kg/dose for female rats and 6 mg/kg/dose for male rats (*Figure 1—source data 1*). Consistent with the MTD of CK21 being different for male/female rats, we observed comparable efficacy of CK21 at 3 mg/kg in female mice (*Figure 2c*), and at 1.5 mg/kg in male mice (*Figure 2—figure supplement 4*). Whether these sex difference in triptolide metabolism will affect dosing in the clinic will have to be investigated in Phase I clinical trials. Nevertheless, despite sex difference, stable exposure of triptolide upon conversion from CK21 resulted in significantly mitigated toxicity, ompared to other analogs such as F60008 that showed a steep release of triptolide which, we speculate, would lead to triptolide overexposure and severe toxicity observed in Phase 1 trials (*Kitzen et al., 2009*). Another triptolide analog, MRx102 had a MTD of 3 mg/kg/dose for the female rats and 4.5 mg/kg/dose for the male rats (*Fidler et al., 2014*). Thus, under the pharmacokinetic profile of CK21, we were able to dose the female athymic

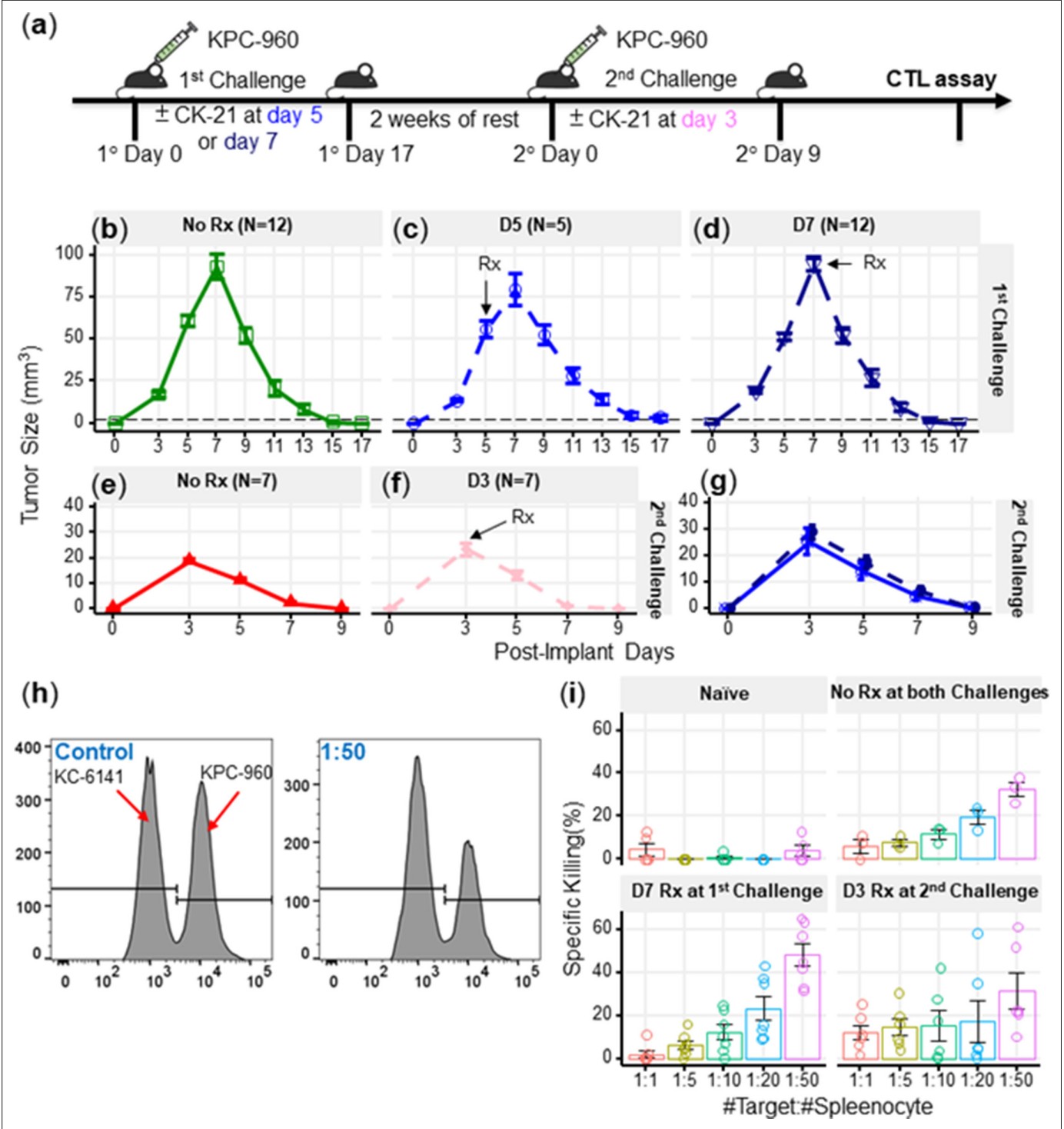

**Figure 7.** CK21 does not exhibit significant immunosuppression in a spontaneous tumor rejection model. (a) Scheme of a subcutaneous model of mouse pancreatic tumor, KPC-960, with CK21 treatment. CK21 was provided at 3 mg/kg daily starting on day 5 or day 7. During secondary challenge, CK21 was provided at 3 mg/kg daily from day 3 post-tumor implantation. Tumor size of mice receiving first challenge (b) without any CK21, (c) with CK21 starting on day 5, (d) or day 7. Tumor size of mice receiving a second challenge (e) without any CK21, or (f) with CK21 treatment starting on day 3. (g) Mice that cleared KPC-960 tumor in (c) and (d) received a second tumor challenge without any CK21; tumor size were quantified weekly (h) Flow plots of CTL assay, another mouse pancreatic tumor, KC-6141, was used as a non-specific target. Quantification of the recovered KPC-960 compared to KC-6141, as a quantification of specific cytotoxic T cell (CTL) killing. (i) Specific CTL killing of KPC-960 cells with splenocytes from (e), (f), (g). Splenocytes

*Figure 7 continued on next page*

*Figure 7 continued*

from naïve mice was included as a negative control. Data presented in all the graphs are mean ± standard error. Statistical analysis: Two-way ANOVA with post-hoc comparison of the means of each time point was conducted for (**b**) and (**e**), (* $P<0.05$, ** $P<0.01$, *** $P<0.001$).

The online version of this article includes the following figure supplement(s) for figure 7:

**Figure supplement 1.** Tumor size of KPC-960 after subcutaneous implantation in nude mice.

nude mice up to 5 mg/kg/day for 28 days with tolerable weight loss (*Figure 2d*), and at 3 mg/kg/day, where CK21 showed potent efficacy and no obvious toxicity (*Figure 2c–e*).

We used rigorous time-course transcriptomic profiling of pancreatic tumors response to CK21 to identify its mechanism of action on patient-derived pancreatic tumor organoids. Overall, the effect of CK21 corresponded to the major reported anti-tumor functions of triptolide, namely transcription inhibition and apoptosis induction. Triptolide was reported by Tivov et al. to covalently bind to XPB, a subunit of the transcription factor TFIIH, resulting in the inhibition of its DNA-dependent ATPase activity, RNA polymerase II (Pol II)-mediated transcription and likely nucleotide excision repair (*Titov et al., 2011*). *Chen et al., 2015* further confirmed that triptolide functioned as a XPB/TFIIH inhibitor to limit promoter-proximal Pol II transcription initiation, resulting in decreased Pol II levels as early as 2 hours of treatment. Likewise, our transcriptome analyses revealed broad downregulation of transcription and transactivation of RNA after 12 hr CK21 treatment (*Figure 5c*). Furthermore, as early as 6 hr of treatment, we observed a significant downregulation of critical transcription factors, including XBP1 and ZNF628 (*Figure 4d*), which may mediate the broad inhibition of RNA and DNA transcription, as well as of RNA transactivation and expression, observed at 12 hr post-CK21 treatment (*Figure 5c*). Inhibition of RNA transcription and blockade of RNA synthesis can potentially lead to programmed cell death. For example, *Santo et al., 2010* used a cyclin-dependent kinase inhibitor to inhibit Pol II phosphorylation and observed induction of apoptosis in myeloma cells. *Cai et al., 2020* also suggested inhibition of Pol II expression and phosphorylation resulted reduced expression of Mcl-1 and X-linked inhibitors of apoptosis (XIAP; ). Similarly, *Carter et al., 2006* reported that tumor cell apoptosis induced by triptolide was accompanied by decrease of XIAP levels. We also observed a significant decrease of XIAP expression after CK21 treatment of two human pancreatic organoids in vitro, and of orthotopically transplanted AsPC-1 tumors in vivo (*Figure 4d, f and g*).

Our analysis of enriched signaling/metabolic pathways (*Figure 5d, e*) predicted the downstream effects of CK21 inhibition of general transcription might lead to tumor cell apoptosis. As a potential consequence of transcription inhibition, genes for the key regulators of NF-κB pathway, such as CHUK, IKBKB and RELA, were significantly downregulated in both organoids (*Figure 6a* and *Figure 6—figure supplement 2*). We also observed decreased p65 expression at a protein level and reduced translocation of the NF-κB complex to the nucleus (*Figure 6b–e*). Therefore, activation of the NF-κB pathway was significantly inhibited after treatment with CK21. In addition to promoting cell proliferation and immune responses (*Park and Hong, 2016*), NF-κB also plays a role in controlling mitochondrial dynamics and cell apoptosis (*Albensi, 2019*). *Pazarentzos et al., 2014* demonstrated the localization of IκBα on the outer membrane of mitochondrial functions to inhibit apoptosis, especially in the tumor cells. *Liu et al., 2004* indicated the inhibition of NF-κB alone can induce the release of cytochrome C from mitochondria (). In our study, we observed a significant downregulation of NFKBIA, which encodes IκBα, in both organoids after CK21 treatment (*Figure 6a* and *Figure 6—figure supplement 2*). In addition, we also observed that the expression of genes encoding five mitochondrial respiratory chain complexes was significantly increased in pancreatic tumor organoids treated with CK21 (*Figure 6f*). Collectively these data suggest a downstream effect of CK21 inhibition of NF-κB is the promotion of dysregulated mitochondrial function and subsequently, increased susceptibility to mitochondrial-mediated intrinsic apoptosis (*Márquez-Jurado et al., 2018*). Nevertheless, we cannot exclude the possibility that the changes in gene expression could reflect different stability of mRNA, and are not directly related to CK21 modifying general transcription.

As upstream regulators, BCL2 family proteins that reside or congregate on the surface of mitochondria govern cell-intrinsic apoptosis (*Adams and Cory, 2001*). BCL2 family proteins have opposing functions in regulating the equilibrium of mitochondrial membrane potential: BCL2 is anti-apoptotic and promotes cell proliferation (*Vaux et al., 1988*), whereas BAX is pro-apoptotic (*Wolter et al., 1997*; *Ly et al., 2003*; *Gross et al., 1999*). Under CK21 treatment, BCL2 expression in pancreatic cancer cells was significantly reduced (*Figure 6h*). Similar observations were reported in leukemic

cells (*Carter et al., 2006*) and melanoma cells (*Tao et al., 2012*) treated with triptolide. Thus CK21 may tip such equilibrium towards permeabilization and release of apoptogenic molecules into cytoplasm (*Gross et al., 1999*). Eventually, effector caspases, such as caspase 3, 6, and 7, are cleaved and activated to induce apoptosis. In our study, we observed a significant increase of cleaved caspase 3 for Panc-1 and both pancreatic tumor organoids (*Figure 6i*). Finally, we noted subtle differences in the extent to which Bcl2 is inhibited and Caspase 3 is activated following CK21 treatment of the two pancreatic tumor cell lines and two patient-derived organoids; these observations underscore the notion that broad inhibition of RNA transcription allows CK21 to leverage distinct vulnerabilities and pathways to achieve apoptosis in different tumor cells.

Taken together, our study describes the development of a novel modified triptolide, CK21, with improved pharmacokinetics, and efficacy for pancreatic tumor cell lines and patient-derived pancreatic tumor organoids. Transcriptomic profiling of the organoids and verification of protein expression collectively point to the induction of tumor cell apoptosis by CK21 is mediated by the inhibition of general transcription, leading to downstream effects involving NF-κB inhibition and mitochondria dysfunction.

## Methods

**Key resources table**

| Reagent type (species) or resource | Designation | Source or reference | Identifiers | Additional information |
|---|---|---|---|---|
| Chemical compound, drug | CK21 | In house | NA | |
| Chemical compound, drug | Gemcitabine | Actavis | 45963-619-59 | |
| Cell line (Homo-sapiens) | AsPC-1 | ATCC | CRL-1682 | |
| Cell line (Homo-sapiens) | Luciferase transfected AsPC-1 | Indiana University | N/A | Luciferase transfected |
| Cell line (Homo-sapiens) | Panc-1 | ATCC | CRL-1469 | |
| Cell line (Mus) | KC-6141 | University of Nebraska | N/A | |
| Cell line (Mus) | KPC-960 | University of Nebraska | N/A | |
| Cell line (Mus) | KPC-961 | University of Nebraska | N/A | |
| Biological sample (Mus) | B6129SF1/J | Jackson Laboratory | 101043 | |
| Biological sample (Mus) | C57BL/6 J | Jackson Laboratory | 000664 | |
| Biological sample (Mus) | Athymic Nude-Foxn1$^{nu}$ | Envigo | | |
| Commercial assay or kit | DMEM | ATCC | 30–2002 | |
| Commercial assay or kit | RPMI | Quality Biological | 112-024-101 | |
| Commercial assay or kit | Fetal bovine serum | Atlanta Biologicals | S115OH | |
| Commercial assay or kit | Penicillin streptomycin | Gibco | 15140–122 | |
| Commercial assay or kit | L-Glutamine | Gibco | 25030–081 | |
| Commercial assay or kit | DMSO | Sigma | 276855 | |
| Commercial assay or kit | Trypsin-EDTA | Stemcell | 07901 | |
| Commercial assay or kit | TrypLE express | Gibco | 12605–010 | |
| Commercial assay or kit | Sodium pyruvate | Gibco | 11360–070 | |
| Commercial assay or kit | MEM nonessential amino acids | Cellgro | 25–025 CL | |
| Commercial assay or kit | 2-Mercaptoethanol | Gibco | 21985–023 | |
| Commercial assay or kit | IntestiCult organoid growth medium | Stemcell | 6005 | |
| Commercial assay or kit | A83-01 | Sigma | SML0788 | |
| Commercial assay or kit | FGF-10 | Sigma | SRP3262 | |

*Continued on next page*

*Continued*

| Reagent type (species) or resource | Designation | Source or reference | Identifiers | Additional information |
|---|---|---|---|---|
| Commercial assay or kit | Gastrin I | Sigma | G9145 | |
| Commercial assay or kit | N-acetylcysteine | Sigma | A9165 | |
| Commercial assay or kit | Nicotinamide | Sigma | N0636 | |
| Commercial assay or kit | B27 supplement | Gibco | 17504–044 | |
| Commercial assay or kit | Primocine | Invivogen | ant-pm-1 | |
| Commercial assay or kit | Y-27632 | Tocris | 1254 | |
| Commercial assay or kit | Matrigel | Corning | 356231 | |
| Commercial assay or kit | TrypLE | Gibco | 12605–010 | |
| Commercial assay or kit | CellTiter 96 AQueous one solution | Promega | G3580 | |
| Commercial assay or kit | Caspase-3/7 green detection | Thermo Fisher | C10427 | |
| Commercial assay or kit | SYTOX dead cell stain | Thermo Fisher | C10427 | |
| Commercial assay or kit | CFSE cell proliferation kit | Thermo Fisher | C34554 | |
| Commercial assay or kit | ACK lysing buffer | Quality Biological | 118-156-101 | |
| Commercial assay or kit | ROS-Glo $H_2O_2$ assay | Promega | G8820 | |
| Commercial assay or kit | NuPAGE 10% Bis-Tris gel | Invitrogen | NP0301BOX | |
| Commercial assay or kit | NuPAGE MES SDS running buffer | Novex | NP002 | |
| Commercial assay or kit | NuPAGE MOPS SDS running buffer | Novex | NP001 | |
| Commercial assay or kit | NuPAGE transfer buffer | Novex | NP0006-1 | |
| Commercial assay or kit | NuPAGE LDS sample reducing agent | Invitrogen | NP0007 | |
| Commercial assay or kit | NuPAGE sample buffer | Invitrogen | NP0009 | |
| Commercial assay or kit | NuPAGE antioxidant | Invitrogen | NP0005 | |
| Commercial assay or kit | TBS Tween-20 buffer | Thermo Scientific | 28360 | |
| Commercial assay or kit | Invitrolon PVDF filter paper | Novex | LC2005 | |
| Commercial assay or kit | PageRuler prestained protein ladder | Thermo Scientific | 26616 | |
| Commercial assay or kit | Methanol | Fisher Scientific | A452-4 | |
| Commercial assay or kit | Pierce protease&phosphatase inhibitor | Thermo Scientific | A32959 | |
| Commercial assay or kit | Bovine serum albumin | Sigma | A7906 | |
| Commercial assay or kit | SuperSignal west pico PLUS | Thermo Scientific | 34579 | |
| Commercial assay or kit | Pierce bradford assay kit | Thermo Scientific | 23246 | |
| Antibody | Anti-beta actin (Rabbit polyclonal) | Abcam | ab8227 | (1:2000) |
| Antibody | Recombinant anti-REDD-1/DDIT4 (Rabbit monoclonal) | Abcam | ab191871 | (1:1000) |
| Antibody | Anti-Caspase-3 (Rabbit polyclonal) | Abcam | ab13847 | (1:500) |
| Antibody | Recombinant anti- BCL2 (Rabbit monoclonal) | Abcam | ab182858 | (1:2000) |
| Antibody | Goat anti-rabbit IgG H&L (Goat polyclonal) | Abcam | ab205718 | (1:10000) |
| Antibody | Phospho-NFkB p65, PE, eBioscience(Mouse monoclonal) | Invitrogen | 12986342 | (1:100) |
| Commercial assay or kit | 4',6-Diamidino-2-Phenylindole, Dilactate | Biolegend | 422801 | (1:1000) |

*Continued on next page*

*Continued*

| Reagent type (species) or resource | Designation | Source or reference | Identifiers | Additional information |
|---|---|---|---|---|
| Commercial assay or kit | PowerUp SYBR green master mix | Applied Biosystem | A25742 | |
| Commercial assay or kit | High capacity cDNA reverse transcription | Applied Biosystem | 4368814 | |
| Commercial assay or kit | D-Luciferin potassium salt | Perkin Elmer | 122799 | |
| Commercial assay or kit | PBS | GenClone | 25–508 | |
| Commercial assay or kit | Cell recovery solution | Corning | 354253 | |
| Commercial assay or kit | RNeasy Plus Mini Kit | Qiangen | 74124 | |
| Commercial assay or kit | DNase I recombinant | Roche | 04536282001 | |

## Study design overview

We synthesized a novel pro-drug of triptolide, CK21, and formulated it into an emulsion. We tested the efficacy of CK21 in vitro using cell proliferation assays and multiple pancreatic cancer cell lines, and in vivo in heterotopic and orthotopic xenograft mouse models. We also tested the efficacy of CK21 against multiple patient-derived pancreatic tumor organoids in vitro and in vivo. We performed transcriptome analysis on the pancreatic organoid response to CK21 in vitro, and on the in vivo response of pancreatic tumors to CK21. This analysis identified the ability of CK21 to reduce overall transcription, inhibit the NF-κB pathway, induce mitochondria dysfunction, and ultimately, mitochondrial-mediated apoptosis. We confirmed inhibition of NF-κB expression and translocation in pancreatic cell lines using imaging flow cytometry, Western blotting and RT-PCR.

## Reagents

Human pancreatic tumor cell lines were obtained from commercial sources. Human tumor organoids were obtained from patients with pancreatic ductal adenocarcinoma, confirmed to be tumor based on pathologic assessment, and developed into organoid culture according to established protocols (*Romero-Calvo et al., 2019*). Luciferase-transfected AsPC-1 tumors (*Shannon et al., 2015*), and mouse tumors from genetically KPC mice that spontaneously develop pancreatic cancer (*Torres et al., 2013*) have been previously described. CK21 was synthesized as described below. All other reagents listed in the Key Resources Table were validated by the manufacturer.

## Synthesis and formulation of CK21

Under nitrogen protection, a mixture of triptolide (1.8 g, 5 mmol) and anhydrous tetrahydrofuran (250 mL) was cooled to –20 °C, and lithium 2,2,6,6-tetramethylpiperidine in tetrahydrofuran/toluene (7.5 mL, 2.0 M, 15 mmol) was added dropwise. After stirring for 30 min, benzoyl chloride (1.05 mL, 7.5 mmol) was added dropwise and reacted for 1 hr, followed again with benzoyl chloride (7.5 mmol) and reacted for another 2 hr. The reaction was quenched by adding aqueous sodium carbonate (10%), and the mixture was extracted with ethyl acetate (250 mL ×3). The organic phases were combined, dried over anhydrous sodium sulfate, and concentrated under reduced pressure. The crude product was separated and purified by silica gel chromatography (dichloromethane: ethyl acetate = 2:1), and the target product (white solid, 2.55 g, yield 90%) was collected and further recrystallized in a mixed organic solvent (dichloromethane/hexane) to obtain a final product (2.13 g, yield 85%, purity >99% by UPLC).

CK21 was dissolved in medium chain triglycerides (MCT) at 90 °C under nitrogen. PC-98T, DSPE-MPEG2000 and glycerol were dissolved in water to form the water phase. The oil phase was dispersed at room temperature in the water phase with high-speed shear mixing (FAS90-22, FLUKO) at 2,800 rpm for 30 min. The pH was adjusted to 4–7, and volume was made up to 100% with water. The final emulsion was obtained by high-pressure homogenization using microfluidizer (M-7125–20 K, MFIC) at 10,000 psi for one cycle and at 18,000 psi for two cycles. Finally, the emulsion was sealed in vials (5 mL: 1.5 mg) after flushing with nitrogen gas and autoclaved at 121 °C for 15 min.

## Characterization of CK21 compound

1 H NMR (Bruker, 400MHz, CDCl3): δ 8.25 (dd, J=1.6 Hz, 8.0 Hz, 2 H), 7.76 (dd, J=1.6 Hz, 8.4 Hz, 2 H), 7.67 (m, 1 H), 7.58 (t, J=7.2 Hz, 2 H), 7.43~7.38 (m, 3 H), 3.80 (d, J=3.2 Hz, 1 H), 3.39 (d, J=2.8 Hz, 1 H), 2.98 (d, J=10 Hz, 1 H), 2.75~2.69 (m, 1 H), 2.63~2.58 (m, 1 H), 2.56 (d, J=6.4 Hz, 1 H), 2.53 (d, J=10 Hz, 1 H), 2.40~2.32 (m, 2 H), 2.21~2.14 (m, 1 H), 1.88 (dd, J=14.0 Hz, 13.2 Hz, 1 H), 1.55~1.52 (m, 1 H), 1.18~1.11 (m, 1 H), 1.15 (s, 3 H), 0.92 (d, J=7.2 Hz, 3 H), 0.82 (d, J=6.8 Hz, 3 H); 13 C NMR (Bruker, 100 MHz, CDCl3): δ168.1, 164.5, 150.3, 142.2, 134.4, 133.5, 131.9, 130.5, 129.9,129.2, 128.9, 128.6, 128.1, 128.0, 72.8, 65.8, 65.3, 60.7, 60.0, 56.5, 53.7, 40.7, 36.7, 29.3, 27.9, 24.6, 17.8, 17.6, 16.7, 15.0.

Mass Spectrometry (AGILENT, ESI+): Calculated for $C_{34}H_{32}O_8$[M]: 568.62, found 569.22 $[M^+H]^+$ and 591.21 $[M^+Na]^+$.

CK21 crystals were obtained by careful evaporation of a mixture of CK21 in combined solvent of dichloromethane and hexane at room temperature. A crystal with size of 0.10×0.03 × 0.02 mm was chosen to be scanned at X-ray diffraction. Data collection was carried out using a Bruker D8 Venture diffractometer with graphite mono-chromated Ga Kα radiation ($\lambda$ =1.34139 Å) at 296 K. Structures were solved by direct methods using the SHELXS program and refined with the SHELXL program (Bruker).

## Pharmacokinetic study of CK21

CK21 emulsion (0.3 mg/mL) was injected intravenously into fasted SD rats at a dose of 3 mg/kg for males and 1.5 mg/kg for females. At designed timepoints, 60 μL blood samples were collected, protein precipitated and centrifuged at 13,000 rpm for 10 min, 4 °C. 5 μL of the supernatant was injected for LC-MS/MS (Q-Trap 6500) analysis. The PK data were calculated using Phoenix WinNonlin 6.3.

## Human pancreatic cancer cell lines and organoids

Human pancreatic cancer cell line, AsPC-1, was cultured in RPMI with 10% fetal bovine serum (FBS), 1% L- Glutamine, and 1% penicillin streptomycin(P/S). Panc-1 was cultured in DMEM with 10% FBS and 1% P/S. Both AsPC-1 and Panc-1 were purchased from ATCC.

Pancreatic tumors from patients with pancreatic ductal adenocarcinoma were collected under IRB12-1108 and IRB13-1149, confirmed to be tumor based on pathologic assessment, and developed into organoid culture according to established protocols (*Romero-Calvo et al., 2019*). Four different organoids, U0118-8, U049MAI, U114SOK, and U123M15-T, were investigated. For the optimal culture, derived organoids were embedded in growth factor reduced Matrigel and cultured in Intesticult complete media, supplemented with A83-01, fibroblast growth factor 10, gastrin I, N-acetyl-L-cysteine, nicotinamide, and B27 supplement, primocin. Tocris Y-27632 dihydrochloride, a selective p160 ROCK inhibitor, was added when thawing the organoids (*Romero-Calvo et al., 2019*).

## In vitro proliferation assay

AsPC-1, Panc-1 and tumor organoids were seeded in 96-well plates and cultured with the indicated concentrations of CK21, or Gemcitabine. CK21 was prepared by dissolving in DMSO and diluting with PBS. At selected times, 20 μL of CellTiter 96 AQueous One solution was added into the 96-well plate, and then incubated at 37 °C for 2 hoursr. The absorbance was read at 490 nm using Spectra Max i3X (Molecular Devices).

## Mice and xenograft

All mouse work that described in this study were approved by the Institutional Animal Care and Use Committee (ACUP72467, ACUP72527). Female or male athymic nude-Foxn1[nu] mice age from 6 to 8 weeks were purchased from Envigo. AsPC-1 or Panc-1 cells were subcutaneously implanted in the scruff of a nude mice at 5×10[6] cells/mice. Mice were treated with different dosages of CK21 daily by intraperitoneal injection. Blank emulsion was provided to the no treatment group. Gemcitabine was also provided to mice at 75 mg/kg once a week as a positive control. The effect of CK21 with another human pancreatic tumor cell line, Panc-1, was also evaluated in the subcutaneous model. The U049MAI organoid was used to test the efficacy of CK21 in the same way.

Tumor size was recorded weekly and calculated by 1/2×L × W (*Kamisawa et al., 2016*). L was the length of the tumor; W was the width of the tumor. Weight of mice were monitored once a week. At the end of the experiment, mice were sacrificed by cervical dislocation. Liver, kidney, pancreas, as well as tumor tissue were harvested and fixed in 10% formalin. Haemotoxylin and Eosin (H&E), terminal deoxynucleotidyl transferase dUTP nick end labeling (TUNEL) staining were performed on respective tissues. All the slides were scanned using ScanScope XT slide scanner and analyzed using Aperio eSlideManager.

### Orthotopic Ttumor Mmodel with Ttransfected AsPC-1

Luciferase-transfected AsPC-1 (*Shannon et al., 2015*) ($1×10^6$ / mouse) was injected into the tail of the pancreas, and one week of tumor implantation, CK21 was provided at 3 mg/kg daily for the treatment group. In the no treatment group, blank emulsion was provided. During the four weeks of treatment, mice were administrated with D-luciferin (Perkin Elmer) and subjected to Xenogen bioluminescence imaging weekly.

### Immunomodulation of CK21 at a spontaneous rejection mice model

Murine pancreatic cancer cell lines were derived from KPC ($Kras^{G12D}$;$Trp53^{R172H}$;$Pdx1$-$Cre$) mice or KC ($Kras^{G12D}$;$Pdx1$-$Cre$) m[i]ice, which spontaneously develop pancreatic cancer (*Torres et al., 2013*). KPC-960 were developed from KPC mice with a mixed background of B6 ×129, and were subcutaneously implanted into female, naïve B6 ×129 mice at $5×10^6$ cells/mice. After spontaneous rejection, mice were rested for 2 weeks and then challenged with KPC-960 cells at $5×10^6$ cells/mice. A dosage of 3 mg/kg of CK21 was provided daily starting at day 5 or day 7. For evaluation of CK21 on memory response, mice that rejected the tumors without any CK21 treatment were rested for 2 weeks and then received a second tumor challenge and 3 mg/kg of CK21 daily, starting at day 3.

Mice that rejected the KPC-960 tumor were sacrificed, splenocytes were collected and ex-vivo specific cytotoxic assay performed. Specifically, target cells KPC-960 and negative control KC-6141 were labeled at 10:1 concentration of carboxyfluorescein succinimidyl ester (CFSE) respectively. Two cell lines were then mix at 1:1 ratio and cultured with harvested splenocytes at 1:1, 1:5, 1:10, 1:20, and 1:50 ratios. After overnight co-culture, cells were subjected to flow cytometry (BD LSR II) to quantify relative cytotoxicity.

### Transcriptome analysis of CK21 treated patient-derived organoids

Two organoids, U049MAI, U123M15-T, were cultured with CK21 at 50 nM for 3 hoursr, 6 hoursr, 9 hoursr, and 12 hoursr. Total RNA was extracted using a RNeasy Plus Mini Kit (Qiagen), and total RNA quantified using the 2100 Bioanalyzer (Agilent). Samples with a RIN >8 was outsourced to Novogene for library construction and sequencing (Illumina Platform (PE150)) with 20 M raw reads/sample. The reads were mapped to the Homosapien genome (GRCh38) using STAR software with ≥95% mapping rate. Differential expression analysis was performed using DESeq2 package in R (*Anders and Huber, 2010*). Molecular and cellular function analysis and pathway enrichment was analyzed using Ingenuine Pathway Analysis software (Qiagen). Duplicate samples were prepared for each condition.

In vivo RNA seq was also performed on orthotropic, luciferase-transfected AsPC-1 tumors. Specifically, luciferase transfected AsPC-1 was implanted into pancreas, and after one week, mice were treated with CK21 at 3 mg/kg for 3 days. Tumor tissues were then resected and RNA seq was performed. Quadruplicate samples were prepared for each condition.

### Imaging Flow cytometry

AsPC-1, Panc-1 were cultured with 50 nM CK21 for 24 hoursr and 48 hoursr. Cells were fixed with 4% paraformaldehyde, and incubated overnight in cocktail of antibody (DPAI, anti-p65) containing 0.1% Triton X-100. Stained cells were subjected to imaging flow cytometry (Amnis ImageStream[X]Mk II) and images analyzed using IDEAS[R] software. Specifically, the 'Similarity' feature in IDEAS[R] indicates the spatial relationship between the p65 and nuclei. Low similarity scores exhibit a predominant cyto-plasmic distribution of p65, whereas high similarity scores indicate a predominant nuclear distribution of p65.

### Western blotting

AsPC-1, Panc-1, U049MAI, or U123M15-T were cultured with 50 nM CK21 for 24 hoursr. Cells then were collected, washed, and lysate for 10 min on ice. Protein concentration of each sample was

detected following the protocol of Pierce Detergent Compatible Bradford Assay. Total of 20 μg denatured protein was then loaded into each lane of NuPAGE Bis-Tris Gel and run using Mini Gel Tank (Invitrogen). Gels were transferred to 0.45 μm Invitrolon PVDF membrane using Mini Blot Module (Invitrogen). Membranes were blocked in 5% BSA overnight at 4 °C. Membranes were then incubated overnight at 4 °C with primary antibodies, including anti-DDIT4, anti-BCL2, anti-Caspase3, or anti-ß-actin. Secondary goat anti-rabbit H&L IgG (HRP) was then incubated for one hour at room temperature. Finally, the chemiluminescent signal was enhanced by with SuperSignal West Pico PLUS Chemiluminescent Substrate, and protein expression was detected using Azure Biosystems 600.

## RT-qPCR

Predesigned primers were purchased from Integrated DNA Technologies, which included XBP1 (Hs.PT.58.1903847), GADD45B (Hs.PT.58.19897476.gs), MYC (Hs.PT.58.26770695), GUSB (Hs.PT.58v.27737538), VAMP1 (Hs.PT.58.26743095), POLR2A (Hs.PT.58.14390640), XIAP (Hs.PT.56a.23056448), DDIT4 (Hs.PT.58.38843854.g), ACTB (Hs.PT.56a.19461448.g) for human tumor organoid samples. DDIT4 (Mm.PT.58.43159110.g), GUSB (Mm.PT.39a.22214848), MYC (Mm.PT.58.13590978), GADD45B (Mm.PT.58.10699383.g), ACTB (Mm.PT.39a.22214843.g), XIAP (Mm.PT.56a.5536843), XBP1 (Mm.PT.58.30961962) for mouse pancreatic tumor cell line samples.

U049MAI or U123M15-T were cultured with 50 nM CK21 for 24 hoursr, total RNA was extracted with an RNeasy Plus Mini Kit (Qiagen) and quantified using Nanodrop 1000 spectrophotometer (Thermo Fisher). RNA of each sample was reverse transcribed into cDNA using High capacity cDNA reverse transcription kit (Applied Biosystems). RT-qPCR were run on QuantStudio 3 (Applied Biosystems) using PowerUp SYBR green master mix with specific primers. RT-qPCR of murine pancreatic cancer cell lines, KC-6141 and KPC-961, were prepared in the same way.

## Cell line authentication

KPC mice cell lines were submitted to ATCC and authenticated using Short Tandem Repeat (STR) analysis as described in the National Institute of Standards and Technology granted U.S. patent (No. 9,556,482). The submitted sample profile is mouse, however a matching reference profile has not previously been established in the ATCC mouse STR database.

Human PDAC cell lines and patient derived organoids were also submitted to ATCC and authenticated using STR analysis as described in ASN-0002-2022. The submitted organoids are confirmed human, but not a match for any profile in the ATCC STR database.The submitted PDAC cell lines are similar to ATCC human cell line: CRL-1682.

## Cell line authentication

U049MAI or U123M15-T were cultured with 50 nM CK21 for 24 hoursr, total RNA was extracted with an RNeasy Plus Mini Kit (Qiagen) and quantified using Nanodrop 1000 spectrophotometer (Thermo Fisher). RNA of each sample was reverse transcribed into cDNA using High capacity cDNA reverse transcription kit (Applied Biosystems). RT-qPCR were run on QuantStudio 3 (Applied Biosystems) using PowerUp SYBR green master mix with specific primers. RT-qPCR of murine pancreatic cancer cell lines, KC-6141 and KPC-961, were prepared in the same way.

## Statistical analysis

Data are presented as means ± standard error (SEM). Statistical analyses were performed using GraphPad Prism software. Differences between groups were analyzed using unpaired t-tests, one-way or two-way ANOVA with post-hoc tests, as indicated in the figure legends.

## Acknowledgements

Data for Figure 1A-G was provided by Drs. Peng Zhang, Bo Qiu, and Fei Xiao from Cinkate Pharmaceutical Corp. All other research was supported in part by a research grant (to University of Chicago) from the Cinkate Pharmaceutical Corp. We thank the Organoid and Primary Culture Research Core at University of Chicago for the gift of patient-derived pancreatic tumor organoids, the Human Tissue

Resource at University of Chicago for tissue processing and staining, the Cytometry and Antibody Technology Core for advising on flow cytometry and the Animal Resources Center at University of Chicago for mouse husbandry services. Dr. Surinder K. Bartra (University of Nebraska Medical Center) provided the mouse pancreatic tumor cell lines. Dr. Barbara Bailey and Dr. Helmut Hanenberg contributed to the generation of the luciferase transfected AsPc-1. We also gratefully acknowledge Dr. Mary Buschman and Ms. Kori Kirby for advising on organoid culture, Stephanie Shen for advising on Western blotting, and Karin Peterson for training on mouse handling.

## Additional information

### Competing interests

Peng Zhang: was an employee of Cinkate Pharmaceutical Corp. Is listed as an inventor on Patent WO2018/019301A1, which covers the design and use of CK21 for pancreatic cancer. Bo Qiu: was an employee of Cinkate Pharmaceutical Corp. Fei Xiao: is listed as an inventor on Patent WO2018/019301A1, which covers the design and use of CK21 for pancreatic cancer. The other authors declare that no competing interests exist.

### Funding

| Funder | Grant reference number | Author |
|---|---|---|
| Cinkate Pharmaceutical Corp | | Anita S Chong |

The funders had no role in study design, data collection and interpretation, or the decision to submit the work for publication.

### Author contributions

Qiaomu Tian, Conceptualization, Data curation, Formal analysis, Validation, Investigation, Methodology, Writing – original draft, Project administration, Writing – review and editing; Peng Zhang, Data curation, Formal analysis, Investigation, Writing – review and editing; Yihan Wang, Youhui Si, Dengping Yin, Investigation; Christopher R Weber, Resources, Formal analysis; Melissa L Fishel, Karen E Pollok, Bo Qiu, Resources; Fei Xiao, Resources, Supervision, Funding acquisition; Anita S Chong, Conceptualization, Formal analysis, Supervision, Funding acquisition, Project administration, Writing – review and editing

### Author ORCIDs

Qiaomu Tian ⓘ https://orcid.org/0000-0003-4267-7362
Anita S Chong ⓘ http://orcid.org/0000-0003-0460-0196

### Ethics

All mouse experiments were approved by the Institutional Animal Care and Use Committee at the University of Chicago, and adhered to the standards of the NIH Guide for the Care and Use of Laboratory Animals. Pancreatic tumors from patients with pancreatic ductal adenocarcinoma were collected under University of Chicago IRB12-1108 and IRB13-1149.

### Decision letter and Author response

Decision letter https://doi.org/10.7554/eLife.85862.sa1
Author response https://doi.org/10.7554/eLife.85862.sa2

## Additional files

### Supplementary files
• MDAR checklist

## Data availability

All data associated with this study are in the article or available at https://doi.org/10.5061/dryad.dbrv15f7s. RNA-seq data are deposited in NCBI GEO under GSE225011.

The following datasets were generated:

| Author(s) | Year | Dataset title | Dataset URL | Database and Identifier |
|---|---|---|---|---|
| Tian Q, Chong A | 2023 | Triptolide analogs induced apoptosis on pancreatic cancer patient-derived organoids | https://www.ncbi.nlm.nih.gov/geo/query/acc.cgi?acc=GSE225011 | NCBI Gene Expression Omnibus, GSE225011 |
| Tian Q, Zhang P, Wang Y, Si Y, Yin D, Weber CR, Fishel ML, Pollok KE, Qiu B, Xiao F, Chong AS | 2023 | A novel triptolide analog downregulates NF-kB and induces mitochondrial apoptosis pathways in human pancreatic cancer | https://doi.org/10.5061/dryad.dbrv15f7s | Dryad, 10.5061/dryad.dbrv15f7s |

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
