## [Editor Report]

Tian et al. describe a novel modified version of the pro-drug triptolide, CK21, and provide evidence for its improved pharmacokinetics and its safety and efficacy in multiple xenograft models of pancreatic cancer. The authors performed transcriptomic analysis upon CK21 treatment which revealed that downregulation of NF-κB and mitochondrial dysfunction induce apoptosis and therefore lead to tumor regression. Downregulation of NF-κB and induction of apoptosis was then validated in vitro and in vivo. These findings have potential clinical significance as the efficacy of CK21 in preclinical PDAC models is compelling.

---

## [Decision Letter]

**Decision letter after peer review:**

Thank you for submitting your article "A novel triptolide analog downregulates NF-κB and induces mitochondrial apoptosis pathways in human pancreatic cancer" for consideration by *eLife*. Your article has been reviewed by 3 peer reviewers, and the evaluation has been overseen by a Reviewing Editor and Wafik El-Deiry as the Senior Editor. The following individual involved in the review of your submission has agreed to reveal their identity: Katerina Gurova (Reviewer #2).

Essential revisions:

*Reviewer #1 (Recommendations for the authors):*

In this manuscript, Tian et al., describe a novel modified version of the pro-drug triptolide, CK21, and provide evidence for its improved pharmacokinetics and its safety and efficacy in multiple xenograft models of pancreatic cancer. The authors performed transcriptomic analysis upon CK21 treatment which revealed that downregulation of NF-κB and mitochondrial dysfunction induce apoptosis and therefore lead to tumor regression. These findings have potential clinical significance as the efficacy of CK21 in preclinical PDAC models is compelling. However, there are also some limitations to their experiments and more validation studies are necessary to strengthen their findings regarding the mechanism of action of the drug.

1) How does the in vivo toxicity and efficacy of CK21 compare to that of other triptolide analogs like minnelide, which is in clinical trials for pancreatic cancers? Side-by-side comparisons are needed to show CK21 is at least as efficacious as other analogs, especially since minnelide was shown to synergize with conventional chemotherapy in PDAC mouse models, whereas CK21 does not appear to.

2) The authors propose that CK21 induce mitochondrial dysfunction based on RNA-seq. They should validate these findings through functional assays such as the Seahorse extracellular flux analyzer.

3) What accounts for the resistance of some PDAC cell lines (such as those derived from the KPC mice) to CK21? The changes in expression of the genes shown in Figure S10 are largely similar to those observed for the sensitive PDAC cells and organoid lines. Transcriptomic analysis of the resistant cells and comparison to the sensitive ones could uncover potential resistance mechanisms and might reveal ways to overcome that resistance. Also, does the in vitro sensitivity, or lack thereof, to CK21 mimic the sensitivity observed in vivo?

4) There are some limitations to the interpretation of the effects of CK21 on the tumor stroma. The authors note that 3D patient derived organoids better recapitulate human PDAC. However, they transplanted these cells subcutaneously instead of orthotopically, which would have had more biological and pharmacological relevance to human disease. Additionally, the limitations of the experiments performed on immunocompetent mice due to the unexplained tumor rejection makes it difficult to interpret the data. No major immunosuppression was observed within the time frame of the treatment. However, the question of whether CK21 is efficacious in syngeneic orthotopic PDAC models is critical and has to be addressed with the right immunocompetent PDAC model, especially since CK21 was shown to have an impact on NF-ΚB which plays a major role in the immune compartment.

5) Given that AsPC-1 cells did not show any cleaved caspase-3/7 in vitro, but TUNEL staining did detect apoptosis in vivo, are there any genes involved in caspase3-independed forms of apoptosis in the in vivo RNA-seq that could explain the observed apoptosis?

6) Since CK21 induced reactive oxygen species and the Ingenuity pathway analysis revealed necrosis as being induced by CK21, was necrosis observed in the tumors following CK21 treatment? This is an important question because necrosis initiates pro-inflammatory signaling cascades whereas apoptosis does not elicit inflammation.

*Reviewer #2 (Recommendations for the authors):*

"CK21 had a T_1/2_ of 1.3 h and 0.225 h for male and female rats respectively. Released triptolide reached Tmax at 0.25 and 0.75 h with a Cmax of 78.3 and 81.9 nM respectively for male and female rats."

These data are confusing:

– Why different doses were chosen for male and female rats?

– How PK parameters can be compared between male and female if they were administered with different doses.

– Variability (error bars or CI) between animals needs to be shown. Now many animals per group were used?

– It is unclear what happened with triptolide at 4 hours? Unmeasured or undetected?

Figure 2. Efficacy studies: What is CK_3_ on figure 2B? All abbreviations need to be explained in figure legend. Why PK was done with IV and efficacy with IP. PK with IP would be much more informative.

"28 days of treatment (Figure 2b). After 28 days of CK21 treatment, no mice demonstrated tumor relapse during the subsequent 6-month follow-up observation (supplement Figure s3)".

Q: all CK21 treated mice or only with certain doses? Lower doses did not show tumor regression.

"To further confirm the lack of toxicity of CK21 (3 mg/kg), we performed HandE and TUNEL staining to detect cell apoptosis on the kidney, liver, and pancreas of mice after 28 days treatment. We observed no toxicity in the kidney, liver, and pancreas tissues after 28 days of treatment (Figure 2e); in contrast, after 14 days of CK21 treatment, AsPC-1 tumors showed a 5-fold increase of TUNEL-positive staining compared to the no Rx group (Figures2f and g)".

To do this comparison TUNEL staining of the kidney, liver, and pancreas tissues and HandE of tumor need to be shown.

How long mice were observed in the orthotopic experiment? Were any of the mice cured? Were they observed for relapse?

Also, the size of error bars for CK21 treated group Figure 2i does not correspond to the difference in the signal shown in Figure 2h

Effect of CK2 on general transcription at therapeutic doses needs to be evaluated? If CK2 inhibits general transcription then all found changes in gene expression possibly reflect different stability of mRNA and are not related to the mechanism of action of CK21.

Figure 4A. Asterisks are confusing, p-value? If yes, how difference between 50nm CK21 and 500nm Gem is significant for U123M15?

Figure 4g does not show a similarity between in vitro and in vivo model response to CK21, just data for AsPC-1. Venn diagram of heat plot may be a better choice.

What is the explanation for the absence of response in KC and KPC mouse models? These abbreviations also need to be explained at first mention.

" The expression of genes encoding five mitochondrial respiratory chain complexes were significantly increased in pancreatic tumor organoids treated with CK21(Figure 6f), consistent with dysregulated mitochondrial function and increased susceptibility to mitochondrial-mediated apoptosis."

This statement does not make sense, since how dysregulation of mitochondrial function may lead to the elevated expression of genes encoding five mitochondrial respiratory chain complexes or vice-versa?

"whether CK21 could prevent the spontaneous regression of KPC-960, CK21 (3 mg/kg daily) therapy was initiated on day 5 or 7 post-implantation. We observed no statistically significant inhibition of tumor regression when CK21 treatment was started on day 5 or 6 post-implantation (Figures7c and d) suggesting that the immunosuppressive activity of CK21 on established primary immune responses is 480 minimal."

In this experiment, you cannot distinguish the reason for tumor regression, immune rejection, or due to CK21 anti-tumor activity. Immune markers in non-tumor-bearing mice need to be measured.

506 "(Figure 1e), and the maximum tolerated dose (MTD) of CK21 was 3 mg/kg/dose for female rats and 6 507 mg/kg/dose for male rats (supplement Figure s13)."

This needs to be mentioned much earlier when PK experiments are described.

*Reviewer #3 (Recommendations for the authors):*

The key limitation of this manuscript is that, beyond an improved formulation of Triptolide, it does not improve our understanding of this agent.

---

## [Author Response]

Essential revisions:Reviewer #1 (Recommendations for the authors):1) How does the in vivo toxicity and efficacy of CK21 compare to that of other triptolide analogs like minnelide, which is in clinical trials for pancreatic cancers? Side-by-side comparisons are needed to show CK21 is at least as efficacious as other analogs, especially since minnelide was shown to synergize with conventional chemotherapy in PDAC mouse models, whereas CK21 does not appear to.

Unfortunately, we did not conduct side-by-side comparison between Minnelide and CK21 in an efficacy study. We agree that this will be an important area of investigation as CK21 moves to clinical trials.

2) The authors propose that CK21 induce mitochondrial dysfunction based on RNA-seq. They should validate these findings through functional assays such as the Seahorse extracellular flux analyzer.

We agree that this would be an important follow-up mechanistic study based on our RNA-seq data to investigate the impact of mitochondrial dysfunction. Unfortunately, we currently do not have the resources to perform these experiments.

3) What accounts for the resistance of some PDAC cell lines (such as those derived from the KPC mice) to CK21? The changes in expression of the genes shown in Figure S10 are largely similar to those observed for the sensitive PDAC cells and organoid lines. Transcriptomic analysis of the resistant cells and comparison to the sensitive ones could uncover potential resistance mechanisms and might reveal ways to overcome that resistance. Also, does the in vitro sensitivity, or lack thereof, to CK21 mimic the sensitivity observed in vivo?

We were not able to identify a single mechanism that explains the relative resistance of some pancreatic cell lines or organoids to CK21. Indeed, we agree that the transcriptomic analysis of both organoids (U049MAI is more sensitive than U123M15-T in vitro) showed similar differentially expressed genes at all time points tested. Likewise, AsPC-1 tumors showed a better sensitivity to CK21 in vivo compared to in vitro results. We speculate that the differential sensitivity might be due to different mutational profiles of the two organoids but their impact was not captured by the 3-12 h transcriptomic analysis and that longer exposure may reveal relevant underlying cause. Furthermore, differences in CK21 metabolism in vivo compared to in vitro may also be relevant.

4) There are some limitations to the interpretation of the effects of CK21 on the tumor stroma. The authors note that 3D patient derived organoids better recapitulate human PDAC. However, they transplanted these cells subcutaneously instead of orthotopically, which would have had more biological and pharmacological relevance to human disease. Additionally, the limitations of the experiments performed on immunocompetent mice due to the unexplained tumor rejection makes it difficult to interpret the data. No major immunosuppression was observed within the time frame of the treatment. However, the question of whether CK21 is efficacious in syngeneic orthotopic PDAC models is critical and has to be addressed with the right immunocompetent PDAC model, especially since CK21 was shown to have an impact on NF-ΚB which plays a major role in the immune compartment.

We agree that orthotopic organoid would be a better model to assess effects of CK21 on the tumor stroma. However, it’s difficult to monitor the growth of the organoids when implanted orthotopically as they are not transfected with luciferase; this, would make the evaluation of the efficacy of CK21 with the orthotopic model very challenging. Hence, we chose to use the subcutaneous model for human organoids in vivo study.

5) Given that AsPC-1 cells did not show any cleaved caspase-3/7 in vitro, but TUNEL staining did detect apoptosis in vivo, are there any genes involved in caspase3-independed forms of apoptosis in the in vivo RNA-seq that could explain the observed apoptosis?

We conducted the caspase-3 assay at 2-24 h of in vitro treatment whereas the in vivo TUNEL study captured the late-stage events (at 2 weeks post-CK21 treatment). Notably, pathway analysis of the statistically significant differentially expressed genes in AsPC-1 tumors after 3 days of CK21 treatment revealed enrichment in Cell Cycle: G2/M DNA Damage Checkpoint Regulation, which corresponds to AsPC-1 tumor apoptosis. We have now included this analysis in Figure 5—figure supplement 2.

6) Since CK21 induced reactive oxygen species and the Ingenuity pathway analysis revealed necrosis as being induced by CK21, was necrosis observed in the tumors following CK21 treatment? This is an important question because necrosis initiates pro-inflammatory signaling cascades whereas apoptosis does not elicit inflammation.

We have assessed the areas of necrosis in the recovered tumors in tumor samples that had been collected at the end of the study (4 weeks of CK21 treatment). A higher percentage necrotic area was observed in the untreated tumors compared to untreated tumors (~0.04% vs. ~0.15%), but the confounding variable was that the tumors from untreated mice were much larger than the CK21 treated tumors, where many had minimally detectable tumors.

Reviewer #2 (Recommendations for the authors):"CK21 had a T_1/2_ of 1.3 h and 0.225 h for male and female rats respectively. Released triptolide reached Tmax at 0.25 and 0.75 h with a Cmax of 78.3 and 81.9 nM respectively for male and female rats."These data are confusing:– Why different doses were chosen for male and female rats?

Gender difference in the toxicity of triptolide in rats has been previously reported and shown to be associated with significantly higher area under the plasma concentration-time curve and peak plasma concentration, lower clearance rate (CL) and longer terminal elimination half-life (t_1/2_) of triptolide in females, and conversely, lower drug exposure levels and greater CL in males. Those studies concluded that gender differential disposition of triptolide may be the cause of increased toxicity in females (Liu et al. 2015). Therefore, our pharmacokinetics study was performed at the highest non-toxic doses for male and female rats.

[1] Liu et al., Gender Differences in the Toxicokinetics of Triptolide after Single- and Multiple-dose Administration in Rats. Drug Res 2015; 65: 602–606

– How PK parameters can be compared between male and female if they were administered with different doses.

Because of different tolerable doses of CK21 in male and female rats and our use of different maximum tolerated doses, we limited our studies to the calculation of T_1/2_, Cmax and the average triptolide concentrations that were maintained for ~2 h post-injection.

– Variability (error bars or CI) between animals needs to be shown. Now many animals per group were used?

We had 3 rats per group and have error bars for Figure 2i, but some are not visible due to low variability. We have included this information in the figure legend.

– It is unclear what happened with triptolide at 4 hours? Unmeasured or undetected?

They are undetected. We have clarified this in the text.

Figure 2. Efficacy studies: What is CK_3_ on figure 2B? All abbreviations need to be explained in figure legend. Why PK was done with IV and efficacy with IP. PK with IP would be much more informative.

We have clarified in the figure legend, and CK_3_ indicates treatment with CK21 at 3 mg/mouse. We chose IP route because it was technically impossible to IV deliver daily for 4 weeks for the efficacy studies; in contrast the PK studies were performed after one CK21 dose and allows us to determine the concentration of CK21 and triptolide in the serum. Unfortunately, we did not perform PK studies with IP administration.

"28 days of treatment (Figure 2b). After 28 days of CK21 treatment, no mice demonstrated tumor relapse during the subsequent 6-month follow-up observation (supplement Figure s3)".Q: all CK21 treated mice or only with certain doses? Lower doses did not show tumor regression.

Tumor-free mice were from CK21 at 3 mg/kg or 5 mg/kg. We have now clarified this in the text.

"To further confirm the lack of toxicity of CK21 (3 mg/kg), we performed Hematoxilin and Eosin (H&E) and TUNEL staining to detect cell apoptosis on the kidney, liver, and pancreas of mice after 28 days treatment. We observed no toxicity in the kidney, liver, and pancreas tissues after 28 days of treatment (Figure 2e); in contrast, after 14 days of CK21 treatment, AsPC-1 tumors showed a 5-fold increase of TUNEL-positive staining compared to the no Rx group (Figures2f and g)".To do this comparison TUNEL staining of the kidney, liver, and pancreas tissues and H&E of tumor need to be shown.

We apologize for this misleading statement. We performed H&E on normal tissues. TUNEL staining on the tumors. We have now clarified this in the text.

How long mice were observed in the orthotopic experiment? Were any of the mice cured? Were they observed for relapse?

In orthotopic experiment, mice were followed up for 3 months. None of these mice were cured and all of them relapsed eventually. We have now included a statement for this model as well.

Also, the size of error bars for CK21 treated group Figure 2i does not correspond to the difference in the signal shown in Figure 2h

Figure 2h are showing bioluminescence images of the tumors of one mouse from D0 to D28 relative to CK21 administration, or untreated mice. Figure 2i quantification of tumor in all the mice in the treated vs non-treated group at the indicated time points. Error bars were calculated as standard error within each group.

Effect of CK2 on general transcription at therapeutic doses needs to be evaluated? If CK2 inhibits general transcription then all found changes in gene expression possibly reflect different stability of mRNA and are not related to the mechanism of action of CK21.

We agree that CK21 ultimately induces inhibition of general transcription as the cell undergoes apoptosis. However, our transcriptional analysis interrogating the effects of CK21 at early 3-12 h timepoints were designed to directly address this concern. Indeed, we observed a significant number of genes that were upregulated, in addition to those that were downregulated, even at the 12 h timepoint (Figure 6a and Figure 6f).

We agree that cannot exclude the possibility that changes in the RNA levels for these genes were due also to changes in the stability of mRNA. We will include this as a potential explanation for the changes in gene transcripts observed.

Figure 4A. Asterisks are confusing, p-value? If yes, how difference between 50nm CK21 and 500nm Gem is significant for U123M15?

The statistical analyses only mapped comparisons between treated groups to No Rx group. Therefore, the asterisks in U123M15 compare either CK21 or GEM treatment to No Rx. p-value ***<0.001; **** p-value < 0.0001. Line indicates the doses that resulted in significant reduction in viability by CK21 or gemcitabine. We have now clarified this in the figure legend.

Figure 4g does not show a similarity between in vitro and in vivo model response to CK21, just data for AsPC-1. Venn diagram of heat plot may be a better choice.

We agree and have replaced Figure 4g with a heatmap of the differentially expressed genes.

What is the explanation for the absence of response in KC and KPC mouse models? These abbreviations also need to be explained at first mention.

Unfortunately, we have not identified a pathway that explains why these genetically-engineered mouse cell lines are resistant to CK21.

We have included an definition of KPC (KrasG12D;Trp53R172H;Pdx1-Cre) or KC (KrasG12D;Pdx1-Cre) mice in the Materials and methods section.

" The expression of genes encoding five mitochondrial respiratory chain complexes were significantly increased in pancreatic tumor organoids treated with CK21(Figure 6f), consistent with dysregulated mitochondrial function and increased susceptibility to mitochondrial-mediated apoptosis."This statement does not make sense, since how dysregulation of mitochondrial function may lead to the elevated expression of genes encoding five mitochondrial respiratory chain complexes or vice-versa?

High expression of mitochondria genes is an indication of apoptotic cells (Marquez-Jurado et al. 2018). We have included this explanation and reference in the manuscript.

Márquez-Jurado, S., Díaz-Colunga, J., das Neves, R.P. et al. Mitochondrial levels determine variability in cell death by modulating apoptotic gene expression. Nat Commun 9, 389 (2018). https://doi.org/10.1038/s41467-017-02787-4

"whether CK21 could prevent the spontaneous regression of KPC-960, CK21 (3 mg/kg daily) therapy was initiated on day 5 or 7 post-implantation. We observed no statistically significant inhibition of tumor regression when CK21 treatment was started on day 5 or 6 post-implantation (Figures7c and d) suggesting that the immunosuppressive activity of CK21 on established primary immune responses is 480 minimal."In this experiment, you cannot distinguish the reason for tumor regression, immune rejection, or due to CK21 anti-tumor activity. Immune markers in non-tumor-bearing mice need to be measured.

We agree that KPC-960 is a complex immune model. We conducted a study of subcutaneous KPC-960 tumor in nude mice and observed limited efficacy by CK21(Figure 7—figure supplement 1). Therefore, we conclude the tumor rejection of this model (KPC-960 in B6X126 mice) is due to immune responses by the host.

506 "(Figure 1e), and the maximum tolerated dose (MTD) of CK21 was 3 mg/kg/dose for female rats and 6 507 mg/kg/dose for male rats (supplement Figure s13)."This needs to be mentioned much earlier when PK experiments are described.

We agree and have included a description in the early of the text (line 122).

Reviewer #3 (Recommendations for the authors):The key limitation of this manuscript is that, beyond an improved formulation of Triptolide, it does not improve our understanding of this agent.

Thank you – we were also disappointed that despite careful transcriptomics analysis at the early time points of the human organoid response to CK21 in vitro, and AsPC-1 model in vivo after 3days of CK21, we were not able to identify a critical pathway or gene that specifies susceptibility to CK21. What we observed was NF-κB activation and upregulation of genes enriched in cell death pathways at the 12 h time point. We therefore conclude that the direct anti-tumor effect of CK21 is cell death through NF-κB activation, mitochondrial dysregulation and ROS production.